# Approximation theory for 1-Lipschitz ResNets

**Davide Murari**
Department of Applied Mathematics and Theoretical Physics
University of Cambridge
dm2011@cam.ac.uk

**Takashi Furuya**
Faculty of Life and Medical Sciences, Department of Biomedical Engineering
Doshisha University
RIKEN AIP
takashi.furuya0101@gmail.com

**Carola-Bibiane Schönlieb**
Department of Applied Mathematics and Theoretical Physics
University of Cambridge
cbs31@cam.ac.uk

## Abstract

1-Lipschitz neural networks are fundamental for generative modelling, inverse problems, and robust classifiers. In this paper, we focus on 1-Lipschitz residual networks (ResNets) based on explicit Euler steps of negative gradient flows and study their approximation capabilities. Leveraging the Restricted Stone–Weierstrass Theorem, we first show that these 1-Lipschitz ResNets are dense in the set of scalar 1-Lipschitz functions on any compact domain when width and depth are allowed to grow. We also show that these networks can exactly represent scalar piecewise affine 1-Lipschitz functions. We then prove a stronger statement: by inserting norm-constrained linear maps between the residual blocks, the same density holds when the hidden width is fixed. Because every layer obeys simple norm constraints, the resulting models can be trained with off-the-shelf optimisers. This paper provides the first universal approximation guarantees for 1-Lipschitz ResNets, laying a rigorous foundation for their practical use.

## 1 Introduction

The flexibility of neural network parameterisations allows them to approximate any regular enough target function arbitrarily well [8, 17, 20, 28, 40, 41]. Despite this desirable aspect, there are several reasons why one would not want a completely unconstrained parameterisation. For instance, unconstrained networks tend to be overly sensitive to input adversarial perturbations because their local Lipschitz constants can be large, making them unreliable classifiers [34]. There are also situations where one is purely interested in modelling specific sets of functions, for example, to turn a constrained optimisation problem into an unconstrained one [19, 31, e.g] A prominent example is the critic in Wasserstein GANs [2], which must be 1-Lipschitz to yield a valid estimate of the 1-Wasserstein distance via the Kantorovich-Rubinstein duality [37]. In this paper, we focus on scalar-valued neural networks that are constrained to be 1-Lipschitz in the Euclidean $\ell^2$ norm. These networks have found extensive applications in inverse problems [16, 32, 33], generative modelling [14, 21, 24], and as a means to improve network resilience to adversarial attacks [23, 29, 33, 35, 36].

39th Conference on Neural Information Processing Systems (NeurIPS 2025).

The approximation properties of constrained networks are poorly understood, and different enforcement strategies can yield markedly different expressiveness. The commonly adopted strategies to constrain a network's Lipschitz constant tend to reduce its expressiveness, leading to noticeable performance drops. This paper studies the approximation properties of Lipschitz-constrained residual networks. We propose new constrained networks that are theoretically able to approximate any scalar 1-Lipschitz function arbitrarily well. Although our analysis is theoretical, the architectures we consider are readily implementable and can be trained like existing Lipschitz-constrained layers.

We next review related work (Section 1.1), summarise our contributions (Section 1.2), and present an outline of the paper (Section 1.3).

## 1.1 Related work

1-**Lipschitz neural networks.** We can identify three principal approaches to enforce or promote Lipschitz constraints: weight normalisation strategies, changes to the network layers, and regularisation strategies. Spectral normalisation or orthogonal weight matrices provide the most well-established constraining procedure for feed-forward neural networks [24, 35]. When working with Residual Neural Networks (ResNets), more changes are necessary due to the skip connection. Results in dynamical systems, numerical analysis, and convex analysis lead to 1-Lipschitz ResNets based on negative gradient flows [23, 33]. These constrained ResNets are the backbone of most of the results in this paper, and we will describe them later. They have also proven to be more efficient than other constraining strategies in terms of performance and computational resource consumption [29]. Because of the empirical decrease in expressiveness with constrained networks, an alternative to hard constraining the Lipschitz constant is regularising the optimisation problem by penalising too large Lipschitz constants [14, 21, 38, 42]. Lipschitz constraints are also considered in more modern architectures, such as Transformers [9, 18].

**Approximation theory for Lipschitz-constrained networks.** Constraining the Lipschitz constant of a network naturally leads to restricted expressiveness, in the sense that a 1-Lipschitz neural network can not approximate $f(x) = x^2$ arbitrarily well, for example. However, a more relevant question is if all the 1-Lipschitz functions can be accurately approximated by a given family of 1-Lipschitz networks. In [1], the authors present the Restricted Stone-Weierstrass Theorem and prove that feed-forward networks based on the $\mathrm{GroupSort}$ activation function and norm-constrained weights are dense in the set of scalar 1-Lipschitz functions. This theorem is fundamental for our derivations as well, and we state it in Section 2. In [25], the authors study feed-forward networks with splines as activation functions and show that they have the same expressiveness as the networks studied in [1]. To the best of our knowledge, no such results are available for 1-Lipschitz ResNets. Closely related to ResNets, we mention [10] where the authors show that Neural ODEs with a constraint on the Lipschitz constant of the flow map are still universal approximators of continuous functions if the linear lifting and projection layers are left unconstrained.

For further motivation of why we study 1-Lipschitz ResNets and why the 1-Lipschitz constraint is relevant, see Appendix F.1.

## 1.2 Main contributions

This paper studies the approximation capabilities of 1-Lipschitz Residual Neural Networks (ResNets). We focus on $\mathrm{ReLU}(x) = \max\{x, 0\}$ as activation function.

Relying on the Restricted Stone-Weierstrass Theorem, in Theorem 3.1 we show that a class of ResNets with residual layers based on explicit Euler steps of negative gradient flows is dense in the set of scalar 1-Lipschitz functions. To achieve this approximation result, we allow for networks of arbitrary width and depth. We also provide an alternative proof by showing, in Theorem 3.2, that this set of networks contains all the 1-Lipschitz piecewise affine functions. This alternative derivation provides further insights into the considered networks. It also informs the other main result of the paper, where the network width is fixed, allowing us to connect these two main theorems. This density result is extended to vector-valued 1-Lipschitz functions as well, see Lemma 5.1.

By interleaving the residual layers with suitably constrained linear maps, we show, in Theorem 4.1, that the width can be fixed while preserving the density in the set of scalar 1-Lipschitz functions. This

second main result allows us to propose a new, practically implementable ResNet-like architecture that is flexible enough to approximate to arbitrary accuracy any scalar 1-Lipschitz function.

We remark that one of the main novelties of our analysis lies in proving the universality of 1-Lipschitz ResNets without relying on the Restricted Stone-Weierstrass Theorem, and in providing two alternative viewpoints to our analysis, thereby gaining a deeper understanding of Lipschitz-continuous ResNets. Not directly relying on the Restricted Stone-Weierstrass Theorem forces us to understand that the set of considered networks contains a very well-studied set of functions: 1-Lipschitz piecewise affine functions. This derivation has significant consequences, since it allows us to transfer the approximation properties of this function class to our neural networks. In other words, part of our theoretical derivations is constructive and is hence informative on what can be represented through the set of networks we consider. On the other hand, the Restrictive-Stone Weierstrass Theorem provides a very general technique for analysing parametric sets of functions and allows for obtaining universality in a non-constructive manner.

### 1.3 Outline of the paper

The paper is structured as follows. Section 2 introduces the primary building block behind our network architectures and some needed notation. In Section 3 we prove that a class of 1-Lipschitz ResNets with an arbitrary number of layers and hidden neurons, is dense in the set of 1-Lipschitz scalar functions. Section 4 provides the second main result, where we show that the same density result can be obtained by fixing the number of hidden neurons and slightly modifying the architecture. Section 5 discusses the implications of these results and outlines potential extensions.

## 2 Preliminaries

In this section, we present the primary building block behind our proposed architectures. We do so after having introduced some necessary notation and definitions.

### 2.1 Notation

We focus on approximating functions in the space

$$\mathcal{C}_1(\mathcal{X}, \mathbb{R}^c) = \left\{ g : \mathcal{X} \to \mathbb{R}^c \;\middle|\; \|g(y) - g(x)\|_2 \le \|y - x\|_2 \; \forall x, y \in \mathcal{X} \right\},$$

where $\mathcal{X} \subseteq \mathbb{R}^d$, $d$ is the input dimension, and $\|x\|_2^2 = x^\top x$ is the Euclidean $\ell^2$-norm. Most of the paper focuses on the case $c = 1$, in which case we write $\mathcal{C}_1(\mathcal{X}, \mathbb{R})$. We will denote the network width with $h$, i.e., the number of hidden neurons, and the network depth with $L$, which is the number of layers. We interchangeably refer to a linear map and its matrix representation with the same notation, e.g., $Q \in \mathbb{R}^{h \times d}$ or $Q : \mathbb{R}^d \to \mathbb{R}^h$. Given a matrix $A \in \mathbb{R}^{r \times s}$, the notation $\|A\|_2$ stands for its spectral norm, i.e., $\|A\|_2 = \sqrt{\lambda_{\max}(A^\top A)}$. We also work with the vector $\ell^1$ norm, which for a vector $x \in \mathbb{R}^d$ is defined as $\|x\|_1 = \sum_{i=1}^d |x_i|$. We write $I_d \in \mathbb{R}^{d \times d}$, $1_d \in \mathbb{R}^d$, $0_{d,h} \in \mathbb{R}^{d \times h}$, and $0_d \in \mathbb{R}^d$ to denote the $d \times d$ identity matrix, a vector of ones, a matrix of zeros, and a vector of zeros, respectively. To refer to the Lipschitz constant of a function $f : \mathbb{R}^d \to \mathbb{R}^c$, we use the notation $\mathrm{Lip}(f)$, i.e., $\|f(y) - f(x)\|_2 \le \mathrm{Lip}(f) \|y - x\|_2$ for any $x, y \in \mathbb{R}^d$.

### 2.2 Universal approximation property

This paper focuses on universal approximation results for $\mathcal{C}_1(\mathcal{X}, \mathbb{R})$, with $\mathcal{X} \subset \mathbb{R}^d$ compact.

**Definition 2.1.** *Let $\mathcal{X} \subset \mathbb{R}^d$ be a compact set, and consider the set of functions $\mathcal{A} \subset \mathcal{C}_1(\mathcal{X}, \mathbb{R})$. We say that $\mathcal{A}$ satisfies the universal approximation property for $\mathcal{C}_1(\mathcal{X}, \mathbb{R})$ if, for any $\varepsilon > 0$ and any $f \in \mathcal{C}_1(\mathcal{X}, \mathbb{R})$ there is a $g \in \mathcal{A}$ such that*

$$\max_{x \in \mathcal{X}} |f(x) - g(x)| < \varepsilon.$$

We now report the statement of the Restricted Stone-Weierstrass Theorem, i.e. [1, Lemma 1], where we adapt the notation and focus on the $\ell^2$-metric, which is the one adopted in our paper.

**Definition 2.2** (Lattice). *Let $\mathcal{X} \subseteq \mathbb{R}^d$ and consider a set $\mathcal{A}$ of functions from $\mathcal{X}$ to $\mathbb{R}$. $\mathcal{A}$ is a lattice if for any pair of functions $f, g \in \mathcal{A}$, the functions $h, k : \mathcal{X} \to \mathbb{R}$ defined as $h(x) = \max\{f(x), g(x)\}$ and $k(x) = \min\{f(x), g(x)\}$ belong to $\mathcal{A}$ as well.*

**Definition 2.3** (Subset separating points). *Let $\mathcal{X} \subseteq \mathbb{R}^d$ be a set with at least two points and consider a set $\mathcal{A} \subset \mathcal{C}_1(\mathcal{X}, \mathbb{R})$. $\mathcal{A}$ separates the points of $\mathcal{X}$ if for any pair of distinct elements $x, y \in \mathcal{X}$ and real numbers $a, b \in \mathbb{R}$ with $|a - b| \leq \|y - x\|_2$, there is an $f \in \mathcal{A}$ such that $f(x) = a$ and $f(y) = b$.*

**Theorem 2.1** (Restricted Stone-Weierstrass). *Let $\mathcal{X} \subset \mathbb{R}^d$ be compact and have at least two points. Let $\mathcal{A} \subset \mathcal{C}_1(\mathcal{X}, \mathbb{R})$ be a lattice separating the points of $\mathcal{X}$. Then $\mathcal{A}$ satisfies the universal approximation property for $\mathcal{C}_1(\mathcal{X}, \mathbb{R})$.*

## 2.3 $1$-Lipschitz residual layers

The main building block of ResNets are layers of the form $x \mapsto x + F_{\theta_\ell}(x) =: \Phi_{\theta_\ell}(x)$, where $x \in \mathbb{R}^d$, and $F_{\theta_\ell} : \mathbb{R}^d \to \mathbb{R}^d$ is a parametric map depending on the parameters collected in $\theta_\ell$. We recall that given two Lipschitz continuous functions $f : \mathbb{R}^{d_1} \to \mathbb{R}^{d_2}$ and $g : \mathbb{R}^{d_2} \to \mathbb{R}^{d_3}$, the Lipschitz constant of $h = g \circ f : \mathbb{R}^{d_1} \to \mathbb{R}^{d_3}$ satisfies $\mathrm{Lip}(h) \leq \mathrm{Lip}(f)\mathrm{Lip}(g)$. For this reason, to build $1$-Lipschitz networks, one typically works with layers that are all $1$-Lipschitz. For a generic Lipschitz continuous function $F_{\theta_\ell}$, it is challenging to have a better bound than $\mathrm{Lip}(\Phi_{\theta_\ell}) \leq 1 + \mathrm{Lip}(F_{\theta_\ell})$, which is the one following from the triangular inequality. However, by making further assumptions of the form of $F_{\theta_\ell}$, it is possible to get $1$-Lipschitz residual layers, as formalised in the following proposition.

**Proposition 2.1** (Theorem 2.3 and Lemma 2.5 in [33]). *Assume $\sigma : \mathbb{R} \to \mathbb{R}$ is $1-$Lipschitz continuous and non-decreasing, and define $\Phi_{\theta_\ell} : \mathbb{R}^h \to \mathbb{R}^h$ as*

$$\Phi_{\theta_\ell}(x) = x - \tau_\ell W_\ell^\top \sigma(W_\ell x + b_\ell) \tag{1}$$

*with $W_\ell \in \mathbb{R}^{h_\ell \times h}$, $\tau_\ell \in \mathbb{R}$, $b \in \mathbb{R}^{h_\ell}$ having $0 \leq \tau_\ell \leq 2/\|W_\ell\|_2^2$. Then, $\mathrm{Lip}(\Phi_{\theta_\ell}) \leq 1$.*

The map $\Phi_{\theta_\ell}$ can be interpreted as a single explicit Euler step of size $\tau_\ell$ for the negative gradient flow differential equation $\dot{x} = -W_\ell^\top \sigma(W_\ell x + b_\ell) = -\nabla g_\ell(x)$, $g_\ell(x) = 1_{h_\ell}^\top \gamma(W_\ell x + b_\ell)$, $\gamma : \mathbb{R} \to \mathbb{R}$ defined by $\gamma' = \sigma$. We define the set of residual layers satisfying the assumptions of Proposition 2.1 and having weights with spectral norm bounded by one:

$$\mathcal{E}_{h,\sigma} = \Big\{ \Phi_\theta : \mathbb{R}^h \to \mathbb{R}^h \ \Big| \ \Phi_\theta(x) = x - \tau W^\top \sigma(Wx + b), \ W \in \mathbb{R}^{k \times h}, \ b \in \mathbb{R}^k,$$

$$\theta = (W, b), \ 0 \leq \tau \leq 2, \ \|W\|_2 \leq 1, \ k \in \mathbb{N} \Big\}.$$

ResNets with layers as in (1) have been used in [23, 29, 33] to improve the robustness to adversarial attacks, and in [33] to approximate the proximal operator and develop a provably convergent Plug-and-Play algorithm for inverse problems.

We will work with residual layers that satisfy the assumptions of Proposition 2.1 and combine them with suitably constrained affine maps to prove our density results. Our focus is on the activation function $\sigma(x) = \mathrm{ReLU}(x) = \max\{0, x\}$, which satisfies the assumptions and simplifies several derivations since it allows us to represent the identity, and the entrywise maximum and minimum functions exactly, which are fundamental operations for our theory. They can be recovered as

$$x = \sigma(x) - \sigma(-x), \ \max\{x, y\} = x + \sigma(y - x), \ \min\{x, y\} = x - \sigma(x - y), \ \forall x, y \in \mathbb{R}^d.$$

Some linear maps belong to $\mathcal{E}_{h,\sigma}$ as well, as formalised in the next lemma.

**Lemma 2.1.** *Let $M \in \mathbb{R}^{h \times h}$ be a symmetric matrix with eigenvalues all in the interval $[0, 1]$. Then the linear map $x \mapsto Mx$ belongs to $\mathcal{E}_{h,\sigma}$ if $\sigma = \mathrm{ReLU}$.*

*Proof.* The matrix $M - I_h$ is symmetric and negative semi-definite. Thus, it can be diagonalised as $M - I_h = -R^\top \Lambda^2 R$ with $R^\top R = RR^\top = I_h$ and $\Lambda = \mathrm{diag}(\lambda_1, ..., \lambda_h)$ having $\|\Lambda\|_2 \leq 1$. Define $V = \Lambda R$. Then,

$$Mx - x = -V^\top(Vx) = -V^\top(\sigma(Vx) - \sigma(-Vx)) = -2\left(\tfrac{1}{\sqrt{2}}V^\top \quad -\tfrac{1}{\sqrt{2}}V^\top\right)\sigma\left(\left(\begin{matrix}\tfrac{1}{\sqrt{2}}V \\ -\tfrac{1}{\sqrt{2}}V\end{matrix}\right)x\right)$$

$$=: -2W^\top\sigma(Wx), \ W^\top = \left(\tfrac{1}{\sqrt{2}}V^\top \quad -\tfrac{1}{\sqrt{2}}V^\top\right),$$

where we used the positive homogeneity of $\sigma$, i.e., $\sigma(\gamma x) = \gamma \sigma(x)$ for all $x \in \mathbb{R}$ and $\gamma \geq 0$. We conclude the desired result by setting $\tau = 2$, and noticing that $\|W\|_2 \leq 1$ since $\|V\|_2 \leq 1$. $\square$

# 3 Density with unbounded width and depth

In this section, we consider the following set of parametric maps

$$\mathcal{G}_{d,\sigma}(\mathcal{X},\mathbb{R}) := \mathcal{C}_1(\mathcal{X},\mathbb{R}) \cap \left\{ v^\top \circ \Phi_{\theta_L} \circ \cdots \circ \Phi_{\theta_1} \circ Q : \mathcal{X} \to \mathbb{R} \; \middle| \; Q(x) = \widehat{Q}x + \widehat{q}, \; \widehat{Q} \in \mathbb{R}^{h \times d}, \right.$$

$$\left. \widehat{q} \in \mathbb{R}^h, \; v \in \mathbb{R}^h, \; \|v\|_2 = 1, \; \Phi_{\theta_\ell} \in \mathcal{E}_{h,\sigma}, \; L, h \in \mathbb{N} \right\}.$$

We remark that in the definition of $\mathcal{G}_{d,\sigma}(\mathcal{X},\mathbb{R})$, the matrix $\widehat{Q}$ is not directly constrained in its norm. However, the intersection with $\mathcal{C}_1(\mathcal{X},\mathbb{R})$ only allows us to consider 1-Lipschitz maps. The lack of explicit constraints over $\widehat{Q}$ leads to problems when implementing these networks, if the goal is to guarantee their 1-Lipschitz regularity. This situation will be resolved by the practicality of the set of networks considered in Section 4. Still, one way to leverage the theory we develop for $\mathcal{G}_{d,\sigma}(\mathcal{X},\mathbb{R})$ in numerical simulations is to leave $\widehat{Q}$ unconstrained while training the model, but simultaneously regularising the loss function so that the Lipschitz constant of the network is controlled by one, as done, for example, in [21].

**Theorem 3.1.** *Let $d \in \mathbb{N}$, $\sigma = \mathrm{ReLU}$ and $\mathcal{X} \subset \mathbb{R}^d$ be compact. Then, $\mathcal{G}_{d,\sigma}(\mathcal{X},\mathbb{R})$ satisfies the universal approximation property for $\mathcal{C}_1(\mathcal{X},\mathbb{R})$.*

We prove this theorem in two ways since they provide different perspectives towards the set $\mathcal{G}_{d,\sigma}(\mathcal{X},\mathbb{R})$. First, in Section 3.1, we verify that $\mathcal{G}_{d,\sigma}(\mathcal{X},\mathbb{R})$ satisfies the assumptions of Theorem 2.1. Then, in Section 3.2, we show that all the piecewise-linear 1-Lipschitz functions from $\mathcal{X}$ to $\mathbb{R}$ belong to $\mathcal{G}_{d,\sigma}(\mathcal{X},\mathbb{R})$.

## 3.1 Proof of Theorem 3.1 based on Restricted Stone-Weierstrass

**Lemma 3.1.** *Let $d \in \mathbb{N}$, $\mathcal{X} \subseteq \mathbb{R}^d$ have at least two points, and $\sigma = \mathrm{ReLU}$. Then $\mathcal{G}_{d,\sigma}(\mathcal{X},\mathbb{R})$ separates the points of $\mathcal{X}$.*

The proof of this lemma is in Appendix A.1, and relies on the fact that all the affine 1-Lipschitz functions from $\mathcal{X}$ to $\mathbb{R}$ belong to $\mathcal{G}_{d,\sigma}(\mathcal{X},\mathbb{R})$.

**Lemma 3.2.** *Let $d \in \mathbb{N}$, $\mathcal{X} \subseteq \mathbb{R}^d$, $\sigma = \mathrm{ReLU}$. Consider two functions $f, g \in \mathcal{G}_{d,\sigma}(\mathcal{X},\mathbb{R})$. There exist $L \in \mathbb{N}$, $h_1, h_2 \in \mathbb{N}$, $v_1 \in \mathbb{R}^{h_1}, v_2 \in \mathbb{R}^{h_2}$ with $\|v_1\|_2 = \|v_2\|_2 = 1$, $Q_1 : \mathbb{R}^d \to \mathbb{R}^{h_1}$ and $Q_2 : \mathbb{R}^d \to \mathbb{R}^{h_2}$ affine maps, $\Phi_{\theta_1}, ..., \Phi_{\theta_L} \in \mathcal{E}_{h_1 + h_2, \sigma}$, and $M \in \mathbb{R}^{(h_1+h_2) \times (h_1+h_2)}$ symmetric positive semi-definite with $\|M\|_2 \leq 1$, such that*

$$\begin{bmatrix} f(x)v_1 \\ g(x)v_2 \end{bmatrix} = M \circ \Phi_{\theta_L} \circ ... \circ \Phi_{\theta_1} \circ \begin{bmatrix} Q_1 \\ Q_2 \end{bmatrix} x.$$

The proof of this lemma is in Appendix A.1, and is based on the fact that the identity map on $\mathbb{R}^h$ belongs to $\mathcal{E}_{h,\sigma}$. We remark that, by Lemma 2.1, the linear map defined by $M$ belongs to $\mathcal{E}_{h_1+h_2,\sigma}$.

**Lemma 3.3.** *Let $d \in \mathbb{N}$, $\mathcal{X} \subseteq \mathbb{R}^d$, and $\sigma = \mathrm{ReLU}$. The set $\mathcal{G}_{d,\sigma}(\mathcal{X},\mathbb{R})$ is a lattice.*

*Proof.* Let $f, g \in \mathcal{G}_{d,\sigma}(\mathcal{X},\mathbb{R})$. We show that $h : \mathcal{X} \to \mathbb{R}$ defined by $h(x) = \max\{f(x), g(x)\}$ belongs to $\mathcal{G}_{d,\sigma}(\mathcal{X},\mathbb{R})$ as well. Analogously, one can show that also $k(x) = \min\{f(x), g(x)\}$ belongs to the set, which is hence a lattice. Recall that since $f$ and $g$ are 1-Lipschitz, $k$ and $h$ will be 1-Lipschitz as well. Thus, we just have to check that $h$ and $k$ can be written as an element of the parametric set we intersect with $\mathcal{C}_1(\mathcal{X},\mathbb{R})$. Lemma 3.2 allows us to write

$$\begin{bmatrix} f(x)v_1 \\ g(x)v_2 \end{bmatrix} = \Phi_{\theta_{L+1}} \circ \Phi_{\theta_L} \circ ... \circ \Phi_{\theta_1} \circ \begin{bmatrix} Q_1 \\ Q_2 \end{bmatrix} x.$$

We then define

$$v = \begin{bmatrix} v_1 \\ 0_{h_2} \end{bmatrix}, \; W_{L+2} = \frac{1}{\sqrt{2}} \begin{bmatrix} -v_1^\top & v_2^\top \end{bmatrix} \in \mathbb{R}^{1 \times (h_1+h_2)}, \; b_{L+2} = 0_{h_1+h_2}, \; \tau_{L+2} = 2.$$

We see that $\|W_{L+2}\|_2 \leq 1$, and hence $\Phi_{\theta_{L+2}}(x) = x - \tau_{L+2} W_{L+2}^\top \sigma(W_{L+2} x + b_{L+2})$ belongs to $\mathcal{E}_{h_1+h_2,\sigma}$. Furthermore, we also notice that $\|v\|_2 = 1$, and that

$$v^\top \circ \Phi_{\theta_{L+2}} \circ \Phi_{\theta_{L+1}} \circ \Phi_{\theta_L} \circ ... \circ \Phi_{\theta_1} \circ \begin{bmatrix} Q_1 \\ Q_2 \end{bmatrix} x = v^\top \circ \Phi_{\theta_{L+2}} \left( \begin{bmatrix} f(x)v_1 \\ g(x)v_2 \end{bmatrix} \right)$$

$$= v^\top \left( \begin{bmatrix} f(x)v_1 \\ g(x)v_2 \end{bmatrix} + \begin{bmatrix} v_1 \\ -v_2 \end{bmatrix} \sigma(g(x) - f(x)) \right) = \begin{bmatrix} v_1^\top 0_{h_2}^\top \end{bmatrix} \begin{bmatrix} v_1 \max\{f(x), g(x)\} \\ v_2 \min\{f(x), g(x)\} \end{bmatrix} = h(x)$$

as desired. $\qquad\square$

We have now proved that $\mathcal{G}_{d,\sigma}(\mathcal{X}, \mathbb{R})$ is a lattice that separates the points of $\mathcal{X}$. Thus, it satisfies the universal approximation property for $\mathcal{C}_1(\mathcal{X}, \mathbb{R})$.

### 3.2 Proof of Theorem 3.1 based on piecewise affine functions

We now present a more constructive reasoning to prove Theorem 3.1. This argument is based on showing that all the scalar, piecewise affine, 1-Lipschitz functions over $\mathcal{X}$ belong to $\mathcal{G}_{d,\sigma}(\mathcal{X}, \mathbb{R})$.

**Definition 3.1.** *A continuous function $f : \mathbb{R}^d \to \mathbb{R}$ is piecewise affine if there exists a finite collection $\mathcal{P}$ of open, pairwise disjoint, connected sets of $\mathbb{R}^d$ with $\mathbb{R}^d = \cup_{P \in \mathcal{P}} \overline{P}$ and $f|_P : P \to \mathbb{R}$ is affine for every $P \in \mathcal{P}$.*

**Remark 3.1.** *Let $\mathcal{X} \subset \mathbb{R}^d$ be compact. We say that a function $g : \mathcal{X} \to \mathbb{R}$ is piecewise affine if there exists a continuous piecewise affine function $\tilde{g} : \mathbb{R}^d \to \mathbb{R}$ such that $g = \tilde{g}|_X$. We say $g : \mathcal{X} \to \mathbb{R}$ is piecewise affine and 1-Lipschitz (on $\mathcal{X}$) if $g = \tilde{g}|_{\mathcal{X}}$ for a piecewise affine 1-Lipschitz function $\tilde{g} : \mathbb{R}^d \to \mathbb{R}$.*

**Lemma 3.4** (Theorem 4.1 in [27]). *Let $f : \mathbb{R}^d \to \mathbb{R}$ be a continuous piecewise affine function. Then, there exists a choice of scalars $b_{i,j} \in \mathbb{R}$ and vectors $a_{i,j} \in \mathbb{R}^d$ such that*

$$f(x) = \max\{f_1(x), ..., f_k(x)\}, \ \ f_i(x) = \min\{a_{i,1}^\top x + b_{i,1}, ..., a_{i,l_i}^\top x + b_{i,l_i}\}. \tag{2}$$

We remark that if $f : \mathbb{R}^d \to \mathbb{R}$ is a continuous piecewise affine 1-Lipschitz function, then necessarily the vectors $a_{i,j} \in \mathbb{R}^d$ appearing in (2) satisfy $\|a_{i,j}\|_2 \leq 1$. This is a consequence of Rademacher's Theorem [13, Theorem 3.1.6], ensuring the almost everywhere differentiability of $f$, which implies that $\|\nabla f(x)\|_2 \leq 1$ for almost every $x \in \mathbb{R}^d$.

We introduce a few fundamental results needed for such a constructive proof.

**Proposition 3.1.** *The functions $\mathbb{R}^d \ni x \mapsto \max\{x_1, ..., x_d\} = f(x) \in \mathbb{R}$ and $\mathbb{R}^d \ni x \mapsto \min\{x_1, ..., x_d\} = g(x) \in \mathbb{R}$ belong to $\mathcal{G}_{d,\sigma}(\mathbb{R}^d, \mathbb{R})$ with $\sigma = \mathrm{ReLU}$.*

*Proof.* We focus on $f$, and the reasoning for $g$ is analogous. The map

$$x \mapsto [\max\{x_1, x_2\} \quad \min\{x_1, x_2\} \quad x_3 \quad \ldots \quad x_d]^\top \tag{3}$$

can be realised as $\Phi_{\theta_1}(x) = x - 2W_1^\top \sigma(W_1 x)$, where

$$W_1 = \begin{bmatrix} -1/\sqrt{2} & 1/\sqrt{2} & 0 & \ldots & 0 \end{bmatrix} \in \mathbb{R}^{1 \times d}.$$

Given that $\max\{x_1, x_2, x_3\} = \max\{\max\{x_1, x_2\}, x_3\}$, it follows that choosing

$$W_2 = \begin{bmatrix} -1/\sqrt{2} & 0 & 1/\sqrt{2} & 0 & \ldots & 0 \end{bmatrix} \in \mathbb{R}^{1 \times d},$$

one has that $\Phi_{\theta_1}(x) - 2W_2^\top \sigma(W_2 \Phi_{\theta_1}(x))$ takes the form

$$[\max\{x_1, x_2, x_3\} \quad \min\{x_1, x_2\} \quad \min\{x_3, \max\{x_1, x_2\}\} \quad x_4 \quad \ldots \quad x_d]^\top.$$

We can thus call $\Phi_{\theta_2}(x) = x - 2W_2^\top \sigma(W_2 x)$. The argument continues up to when, setting $v = e_1$, the first vector of the canonical basis of $\mathbb{R}^d$, we get

$$v^\top \circ \Phi_{\theta_{d-1}} \circ ... \circ \Phi_{\theta_1}(x) = \max\{x_1, ..., x_d\} = f(x)$$

as desired. We remark that, in this case, $Q = I_d$ and $h = d$. $\qquad\square$

This analysis, together with Proposition 3.1, implies that scalar, piecewise affine, $1-$Lipschitz functions all belong to $\mathcal{G}_{d,\sigma}(\mathbb{R}^d, \mathbb{R})$, as we formalise in the following theorem.

**Theorem 3.2.** *Any piecewise affine 1-Lipschitz function $f : \mathbb{R}^d \to \mathbb{R}$ can be represented by a network in $\mathcal{G}_{d,\sigma}(\mathbb{R}^d, \mathbb{R})$ with $\sigma = \mathrm{ReLU}$.*

**Remark 3.2.** *Let $\mathcal{X} \subset \mathbb{R}^d$ be compact and $g : \mathcal{X} \to \mathbb{R}$ a piecewise affine 1-Lipschitz function on $\mathcal{X}$ in the sense of Remark 3.1. Then, there exists a piecewise affine 1-Lipschitz function $\tilde{g} : \mathbb{R}^d \to \mathbb{R}$ such that $g = \tilde{g}|_{\mathcal{X}}$. Applying Theorem 3.2 to $\tilde{g}$ yields a network $\mathcal{N} \in \mathcal{G}_{d,\sigma}(\mathbb{R}^d, \mathbb{R})$ such that $\mathcal{N} = \tilde{g}$ on $\mathbb{R}^d$; hence, by restriction, $\mathcal{N}|_{\mathcal{X}} \in \mathcal{G}_{d,\sigma}(\mathcal{X}, \mathbb{R})$ and $\mathcal{N}|_{\mathcal{X}} = g$ on $\mathcal{X}$.*

See Appendix A.2 for the proof. A proof of Theorem 3.1 then follows from the universal approximation property of piecewise affine 1-Lipschitz maps defined on a compact set $\mathcal{X}$ in $\mathcal{C}_1(\mathcal{X}, \mathbb{R})$.

**Lemma 3.5.** *The set of piecewise affine 1-Lipschitz functions over $\mathcal{X} \subset \mathbb{R}^d$, a compact set, satisfies the universal approximation property for $\mathcal{C}_1(\mathcal{X}, \mathbb{R})$.*

To prove this lemma, one could use the Restricted Stone-Weierstrass Theorem, since the set of piecewise affine 1-Lipschitz functions is a lattice separating points. We provide a more explicit and direct proof of Lemma 3.5 for the case $\mathcal{X}$ is a convex polytope in Appendix A.2.

## 4 Density with fixed width and unbounded depth

Theorem 3.1 ensures that it is possible to approximate to arbitrary accuracy any scalar 1-Lipschitz function over a compact set $\mathcal{X}$ by using ResNets relying on negative gradient steps. This result is informative but it has two drawbacks: (i) the elements of $\mathcal{G}_{d,\sigma}(\mathcal{X}, \mathbb{R})$ do not have explicit constraints on the affine lifting layer $Q$, making them challenging to implement, (ii) there is no control neither on the depth nor on the width of the networks in $\mathcal{G}_{d,\sigma}(\mathcal{X}, \mathbb{R})$. We now address these limitations by providing a second set of 1-Lipschitz networks, which are easier to implement and have fixed width.

For the derivations in this section, we need to introduce two sets of suitably constrained affine maps. Let $k \in \mathbb{N}$, fix a vector $m \in \mathbb{N}^k$, and call $\alpha_m = \|m\|_1 = m_1 + ... + m_k$. We define

$$\widetilde{\mathcal{L}}_m = \left\{ A \in \mathbb{R}^{\alpha_m \times \alpha_m} \ \middle| \ A = \begin{bmatrix} A_{11} & ... & A_{1k} \\ \vdots & \ddots & \vdots \\ A_{k1} & ... & A_{kk} \end{bmatrix}, A_{ij} \in \mathbb{R}^{m_i \times m_j}, \sum_{j=1}^{k} \|A_{ij}\|_2 \leq 1, \ i = 1, ..., k \right\},$$

$$\mathcal{L}_m = \left\{ A : \mathbb{R}^{\alpha_m} \to \mathbb{R}^{\alpha_m} \ \middle| \ \exists \widehat{A} \in \widetilde{\mathcal{L}}_m, \ \widehat{a} \in \mathbb{R}^{\alpha_m} : A(u) = \widehat{A}u + \widehat{a}, \forall u \in \mathbb{R}^{\alpha_m} \right\},$$

$$\widetilde{\mathcal{R}}_{d,m} = \left\{ B \in \mathbb{R}^{\alpha_m \times d} \ \middle| \ B = \begin{bmatrix} B_1 \\ \vdots \\ B_k \end{bmatrix}, B_i \in \mathbb{R}^{m_i \times d}, \|B_i\|_2 \leq 1, \ i = 1, ..., k \right\},$$

$$\mathcal{R}_{d,m} = \left\{ Q : \mathbb{R}^d \to \mathbb{R}^{\alpha_m} \ \middle| \ \exists \widehat{Q} \in \widetilde{\mathcal{R}}_{d,m}, \ \widehat{q} \in \mathbb{R}^{\alpha_m} : Q(x) = \widehat{Q}x + \widehat{q}, \forall x \in \mathbb{R}^d \right\}.$$

We also extend the set of functions $\mathcal{E}_{h,\sigma}$ to a subset of $\mathcal{E}_{h+3,\sigma}$ as follows

$$\widetilde{\mathcal{E}}_{h,\sigma} = \left\{ \Phi_\theta : \mathbb{R}^{h+3} \to \mathbb{R}^{h+3} \ \middle| \ \Phi_\theta(x) = \begin{bmatrix} \max\{x_1, x_2\} \\ \min\{x_1, x_2\} \\ x_3 \\ \widetilde{\Phi}_\theta(x_{4:}) \end{bmatrix}, \ \widetilde{\Phi}_\theta \in \mathcal{E}_{h,\sigma} \right\},$$

where, for $x \in \mathbb{R}^{h+3}$, $x_{4:} \in \mathbb{R}^h$ denotes a vector coinciding with $x$ to which the first three entries are removed. We remark that the first two components of the functions in $\widetilde{\mathcal{E}}_{h,\sigma}$ resemble the MaxMin activation in [1] or the Orthogonal Permutation Linear Unit in [7].

**Lemma 4.1.** *Let $h \in \mathbb{N}$ and $\sigma = \mathrm{ReLU}$. The set $\widetilde{\mathcal{E}}_{h,\sigma}$ is a subset of $\mathcal{E}_{h+3,\sigma}$.*

See Appendix C for the proof.

Fix $h \geq 3$. We now consider the set

$$\widetilde{\mathcal{G}}_{d,\sigma,h}(\mathcal{X}, \mathbb{R}) := \left\{ v^\top \circ \Phi_{\theta_L} \circ A_{L-1} \circ \cdots \circ \Phi_{\theta_2} \circ A_1 \circ \Phi_{\theta_1} \circ Q : \mathcal{X} \to \mathbb{R} \ \middle| \ m = (1, 1, 1, h-3), \right.$$

$$\left. Q \in \mathcal{R}_{d,m}, v \in \mathbb{R}^h, \|v\|_1 \leq 1, A_1, ..., A_{L-1} \in \mathcal{L}_m, \Phi_{\theta_\ell} \in \widetilde{\mathcal{E}}_{h-3,\sigma}, L \in \mathbb{N} \right\}.$$

**Lemma 4.2.** *Let $\sigma = \mathrm{ReLU}$, $d, h \in \mathbb{N}$, with $h \geq 3$. All the functions in $\widetilde{\mathcal{G}}_{d,\sigma,h}(\mathbb{R}^d, \mathbb{R})$ are 1-Lipschitz.*

See Appendix C for the proof. This lemma addresses the first limitation in the definition of $\mathcal{G}_{d,\sigma}(\mathcal{X}, \mathbb{R})$, given that we have explicit constraints over all the terms defining a neural network in $\widetilde{\mathcal{G}}_{d,\sigma,h}(\mathcal{X}, \mathbb{R})$.

We now state the second main result of this paper.

**Theorem 4.1.** *Let $d \in \mathbb{N}$, $\sigma = \mathrm{ReLU}$, and $\mathcal{X} \subset \mathbb{R}^d$ be compact. The set $\widetilde{\mathcal{G}}_{d,\sigma,d+3}(\mathcal{X}, \mathbb{R})$ satisfies the universal approximation property for $\mathcal{C}_1(\mathcal{X}, \mathbb{R})$.*

This result ensures we can fix the network width to $h = d + 3$ and preserve the universal approximation property. We will further comment on the connections between the two sets $\mathcal{G}_{d,\sigma}(\mathcal{X}, \mathbb{R})$ and $\widetilde{\mathcal{G}}_{d,\sigma,h}(\mathcal{X}, \mathbb{R})$ and on extensions of the set $\widetilde{\mathcal{G}}_{d,\sigma,h}(\mathcal{X}, \mathbb{R})$ in Section 5. We remark that our proof of Theorem 4.1 relies on setting $\widetilde{\Phi}_\theta(x_{4:}) = x_{4:}$ for the elements in $\widetilde{\mathcal{E}}_{h,\sigma}$. Still, we present the results for the larger set of allowed residual maps $\widetilde{\mathcal{E}}_{h,\sigma}$ since, in practice, this additional freedom can lead to more efficient approximations of target maps than the one provided constructively in our proof.

### 4.1 Proof of Theorem 4.1

This proof follows similar ideas as the one presented in Section 3.2.

**Proposition 4.1.** *Fix $n \in \mathbb{N}$, $a_1, ..., a_n \in \mathbb{R}^d$, and $b_1, ..., b_n \in \mathbb{R}$, with $\|a_1\|_2, ..., \|a_n\|_2 \leq 1$. The functions $\mathbb{R}^d \ni x \mapsto \max\{a_1^\top x + b_1, ..., a_n^\top x + b_n\} = f(x) \in \mathbb{R}$ and $\mathbb{R}^d \ni x \mapsto \min\{a_1^\top x + b_1, ..., a_n^\top x + b_n\} = g(x) \in \mathbb{R}$ belong to $\widetilde{\mathcal{G}}_{d,\sigma,h}(\mathbb{R}^d, \mathbb{R})$ with $\sigma = \mathrm{ReLU}$ and $h = d + 3$.*

*Proof.* We focus on $f$, and a similar reasoning applies for $g$. Set $\widetilde{W}_1 = 0_{d,d}$, $\widetilde{b}_1 = 0_d$, and $Q$ so that

$$\Phi_{\theta_1} \circ Q(x) = \left[\max\{a_1^\top x + b_1, a_2^\top x + b_2\} \quad \min\{a_1^\top x + b_1, a_2^\top x + b_2\} \quad 0 \quad x^\top\right]^\top,$$

where $\widetilde{\Phi}_{\theta_1}(x) = x - 2\widetilde{W}_1^\top \sigma(\widetilde{W}_1 x + \widetilde{b}_1)$. Set $m = (1, 1, 1, d)$. Let us then introduce $A_1 \in \mathcal{L}_m$ defined as $A_1(x) = \widetilde{A}_1 x + \widetilde{a}_1$ where

$$\widetilde{A}_1 = \begin{bmatrix} 1 & 0 & 0 & 0_d^\top \\ 0 & 0 & 0 & a_3^\top \\ 0 & 0 & 0 & 0_d^\top \\ 0_d & 0_d & 0_d & I_d \end{bmatrix} \in \mathbb{R}^{h \times h}, \quad \widetilde{a}_1 = \begin{bmatrix} 0 \\ b_3 \\ 0 \\ 0 \end{bmatrix} \in \mathbb{R}^h,$$

so that

$$A_1 \circ \Phi_{\theta_1} \circ Q(x) = \left[\max\{a_1^\top x + b_1, a_2^\top x + b_2\} \quad a_3^\top x + b_3 \quad 0 \quad x^\top\right]^\top \in \mathbb{R}^h.$$

The next step is to build $\Phi_{\theta_2}$ so that

$$\Phi_{\theta_2} \circ A_1 \circ \Phi_{\theta_1} \circ Q(x) = \begin{bmatrix} \max\{a_1^\top x + b_1, a_2^\top x + b_2, a_3^\top x + b_3\} \\ \min\{\max\{a_1^\top x + b_1, a_2^\top x + b_2\}, a_3^\top x + b_3\} \\ 0 \\ x \end{bmatrix}.$$

The reasoning extends up to when we reach the final configuration, after $L = n - 1$ residual maps, where $v^\top \circ \Phi_{\theta_L} \circ A_{L-1} \circ \Phi_{\theta_{L-1}} \circ ... \circ A_1 \circ \Phi_{\theta_1} \circ Q(x) = f(x)$, by fixing $v = e_1 \in \mathbb{R}^h$. $\qquad\square$

We remark that in the proof of Proposition 4.1 the third entry is irrelevant, and the fourth component is used as a memory of the original input $x$. The third component is fundamental in the proof of Theorem 4.2, which is why it is included. Keeping track of $x$ is essential to recover the affine pieces $a_i^\top x + b_i$ online, without generating them all at the beginning as we did in the proof of Theorem 3.2. This operation allows us to detach the hidden dimension $h$ from the number of linear pieces, and get a universal approximation theorem for a fixed network width.

**Theorem 4.2.** *Any piecewise affine 1-Lipschitz function $f : \mathbb{R}^d \to \mathbb{R}$ can be represented by a network in $\widetilde{\mathcal{G}}_{d,\sigma,h}(\mathbb{R}^d, \mathbb{R})$ with $\sigma = \mathrm{ReLU}$ and $h = d + 3$.*

The proof of Theorem 4.2 relies on the representation of piecewise affine 1-Lipschitz functions in (2), the construction presented in the proof of Proposition 4.1, and on using the third component as a running maximum of the previously computed minima. See Appendix C for the full proof.

The proof of Theorem 4.1 then follows by combining Theorem 4.2 with Lemma 3.5.

We discuss how the network size depends on the input dimension $d \in \mathbb{N}$ in Appendix F.2.

**Remark 4.1.** *The set of networks $\widetilde{\mathcal{G}}_{d,\mathrm{ReLU},d+3}(\mathcal{X}, \mathbb{R})$ contains all the piecewise affine 1-Lipschitz functions, as stated in Theorem 4.2. On the other hand, it is also true that all the elements of $\widetilde{\mathcal{G}}_{d,\mathrm{ReLU},d+3}(\mathcal{X}, \mathbb{R})$ are piecewise affine 1-Lipschitz functions. The latter result follows from the fact that $\mathrm{ReLU}$ is piecewise affine, and we only compose it with affine maps. This reasoning allows us to conclude that the new architecture that we propose and study in Theorem 4.1 coincides with the set of piecewise affine 1-Lipschitz functions, and provides yet another representation strategy for this lattice of functions.*

# 5 Discussion and future work

We now connect our two main theorems, comment on their practical value, and provide relevant extensions.

**Different proving strategies.** To prove Theorem 3.1, we followed two strategies: verified that the assumptions of the Restricted Stone-Weierstrass Theorem are satisfied by $\mathcal{G}_{d,\sigma}(\mathcal{X}, \mathbb{R})$, and verified that $\mathcal{G}_{d,\sigma}(\mathcal{X}, \mathbb{R})$ contains all the scalar piecewise affine and 1-Lipschitz functions. The two arguments are strictly related. In fact, the first one relies on showing that $\mathcal{G}_{d,\sigma}(\mathcal{X}, \mathbb{R})$ is a lattice separating the points of $\mathcal{X}$, while the second shows that $\mathcal{G}_{d,\sigma}(\mathcal{X}, \mathbb{R})$ contains a lattice that separates points. This latter strategy is the one we used to prove Theorem 4.1 as well.

**Necessity for the affine maps $A_1, ..., A_{L-1} \in \mathcal{L}_m$.** To obtain universality while maintaining the width fixed, we introduced affine layers between the residual gradient steps. Setting them to identity maps would lead to a subset of $\widetilde{\mathcal{G}}_{d,\sigma,h}(\mathcal{X}, \mathbb{R})$ which might not be universal. Since these maps are not 1-Lipschitz as maps from $\mathbb{R}^{\alpha m}$ to itself, we had to restrict the set of allowed gradient steps to $\widetilde{\mathcal{E}}_{h,\sigma} \subset \mathcal{E}_{h+3,\sigma}$. It is thus interesting to further explore if it is possible to remove these affine maps and allow for more general gradient steps. Still, there are two fundamental aspects to mention.

First, to get a 1-Lipschitz network, it is not necessary to have all the layers that are 1-Lipschitz, as demonstrated by the maps in $\widetilde{\mathcal{G}}_{d,\sigma,h}(\mathcal{X}, \mathbb{R})$. This idea is explored in [4], where the authors build 1-Lipschitz networks combining 1-Lipschitz maps as in $\mathcal{E}_{d,\sigma}$ with positive gradient steps of the form $u \mapsto u + \tau W^\top \sigma(Wu + b)$. It will thus be interesting to further explore this path for future research.

Second, the restriction defined in $\widetilde{\mathcal{E}}_{h,\sigma}$ provides a rather minimal set to carry out the proofs in this paper. One can generalise it while preserving the Lipschitz property and potentially getting more practically efficient networks. We provide a generalisation of $\widetilde{\mathcal{E}}_{h,\sigma}$ and $\widetilde{\mathcal{G}}_{d,\sigma,h}(\mathcal{X}, \mathbb{R})$ in Appendix D.

**Connections between Theorem 3.1 and Theorem 4.1.** Both the main results we proved in this paper ensure the universality of neural networks that rely on residual layers coming from negative gradient flows. Theorem 3.1 focuses on architectures studied in [23, 29, 33], while Theorem 4.1 considers a new, practically implementable constrained architecture that we propose. It is essential to note that in [23, 29, 33] the elements in $\mathcal{G}_{d,\sigma}(\mathcal{X}, \mathbb{R})$ are implemented with a unit-norm constraint on the lifting map $Q : \mathbb{R}^d \to \mathbb{R}^h$. Our theory does not allow us to say that this constraining strategy provides a set of networks dense in $\mathcal{C}_1(\mathcal{X}, \mathbb{R})$, and hence leaving it unconstrained while regularising for the network to be almost 1-Lipschitz would be a better strategy according to our analysis.

Theorem 4.1 trades the unlimited width of Theorem 3.1 for additional constrained linear layers. Still, there are several similarities between the two theorems. For example, the constrained maps in $\widetilde{\mathcal{E}}_{h,\sigma}$ appear also in the proof of Theorem 3.1, see, for example, (3).

**Extension to multivalued functions.** This paper focuses on $\mathcal{C}_1(\mathcal{X}, \mathbb{R})$. Some of our results extend to the functions in $\mathcal{C}_1(\mathcal{X}, \mathbb{R}^c)$, as formalised by the following lemma, with proof in Appendix E.

**Lemma 5.1.** *Let $c, d \in \mathbb{N}$, $\mathcal{X} \subset \mathbb{R}^d$ be compact, and $\sigma = \mathrm{ReLU}$. Define the set*

$$\mathcal{G}_{c,d,\sigma}(\mathcal{X}, \mathbb{R}^c) := \left\{ P \circ \Phi_{\theta_L} \circ \cdots \circ \Phi_{\theta_1} \circ Q : \mathcal{X} \to \mathbb{R}^c \; \middle| \; Q(x) = \widehat{Q}x + \widehat{q}, \; \widehat{Q} \in \mathbb{R}^{h \times d}, \right.$$

$$\left. \widehat{q} \in \mathbb{R}^h, \; P \in \mathbb{R}^{c \times h}, \; \|P\|_{2,\infty} = 1, \; \Phi_{\theta_\ell} \in \mathcal{E}_{h,\sigma}, \; L, h \in \mathbb{N} \right\}.$$

*Then, for any $f \in \mathcal{C}_1(\mathcal{X}, \mathbb{R}^c)$ and $\varepsilon > 0$, exists $g \in \mathcal{G}_{c,d,\sigma}(\mathcal{X}, \mathbb{R}^c)$ with $\max_{x \in \mathcal{X}} \|f(x) - g(x)\|_2 \leq \varepsilon$.*

Since we are not intersecting with the set of 1-Lipschitz functions, this time, it is not true that $\mathcal{G}_{c,d,\sigma}(\mathcal{X}, \mathbb{R}^c) \subset \mathcal{C}_1(\mathcal{X}, \mathbb{R}^c)$. In fact, since $\|P\|_{2,\infty} = \max_{i=1,\dots,c} \|e_i^\top P\|_2$, where $e_i \in \mathbb{R}^c$ is the $i$-th vector of the canonical basis, we have $\mathcal{G}_{d,\sigma}(\mathcal{X}, \mathbb{R}) \subset \mathcal{G}_{1,d,\sigma}(\mathcal{X}, \mathbb{R}^1)$. Extending the universality result of $\widetilde{\mathcal{G}}_{d,\sigma,h}(\mathcal{X}, \mathbb{R})$ and enforcing the Lipschitz constraint will be the topic for future research.

**Extension to larger sets of parametric functions.** As for any universal approximation theorem, the theoretical analysis we provide can be extended to sets containing $\mathcal{G}_{d,\sigma}(\mathcal{X}, \mathbb{R})$ and $\widetilde{\mathcal{G}}_{d,\sigma,h}(\mathcal{X}, \mathbb{R})$. For example, the universality of $\mathcal{G}_{d,\sigma}(\mathcal{X}, \mathbb{R})$ implies that the universality persists relaxing the constraint over $v \in \mathbb{R}^h$ from $\|v\|_2 = 1$ to $\|v\|_2 \leq 1$. Similarly, we can remove the constraints on the matrices $W_\ell$, and allow for $0 \leq \tau_\ell \leq 2/\|W_\ell\|_2^2$, still leading to 1-Lipschitz Euler steps, see Proposition 2.1. In fact, $\|W_\ell\|_2 \leq 1$ with $\tau_\ell \in [0, 2]$ is a particular instance of this constraint. Relaxing such constraints could improve the numerical performance, given that the model would be less restricted.

**Extension to other activation functions.** Our proofs strongly rely on the properties of $\mathrm{ReLU}$. The same results could be obtained by any other activation functions that are positively homogeneous, can represent the identity map, and the entrywise maximum and minimum functions. When this is not the case, developing a similar theory would require significantly different arguments. For the further discussion of other activation functions, see Appendix F.3.

**Implementability of our networks.** The elements of $\mathcal{G}_{d,\sigma}(\mathcal{X}, \mathbb{R})$ and $\widetilde{\mathcal{G}}_{d,\sigma,h}(\mathcal{X}, \mathbb{R})$ are neural networks that can be numerically implemented. All the weight constraints can be efficiently enforced in a projected gradient descent fashion. The spectral norms can be estimated via the power method, see [24], whereas row-wise constraints as in $\mathcal{L}_m$ can be enforced by dividing by the $\ell^2$ norms of the rows. We further remark that the lack of explicit constraints on the lifting map $Q$ in $\mathcal{G}_{d,\sigma}(\mathcal{X}, \mathbb{R})$ can be overcome by regularising the loss function with a term penalising the violation of the Lipschitz constraint, such as $\sum_{i=1}^N (\mathrm{ReLU}(\|\nabla_x \mathcal{N}_\theta(x_i)\|_2 - 1))^2$. This additional term promotes the local Lipschitz constant of the network $\mathcal{N}_\theta : \mathbb{R}^d \to \mathbb{R}$ to be smaller or equal than one, see [21]. The locations $x_1, \dots, x_N \in \mathcal{X}$ can be randomly sampled during each training iteration. For some empirical validation of the implementability and trainability of our networks, see Appendix G.

## Acknowledgements

DM acknowledges support from the EPSRC programme grant in 'The Mathematics of Deep Learning', under the project EP/V026259/1. TF was supported by JSPS KAKENHI Grant Number JP24K16949, 25H01453, JST CREST JPMJCR24Q5, JST ASPIRE JPMJAP2329. CBS acknowledges support from the Philip Leverhulme Prize, the Royal Society Wolfson Fellowship, the EPSRC advanced career fellowship EP/V029428/1, the EPSRC programme grant EP/V026259/1, and the EPSRC grants EP/S026045/1 and EP/T003553/1, EP/N014588/1, EP/T017961/1, the Wellcome Innovator Awards 215733/Z/19/Z and 221633/Z/20/Z, the European Union Horizon 2020 research and innovation programme under the Marie Skodowska-Curie grant agreement NoMADS and REMODEL, the Cantab Capital Institute for the Mathematics of Information and the Alan Turing Institute. This research was also supported by the NIHR Cambridge Biomedical Research Centre (NIHR203312). The views expressed are those of the author(s) and not necessarily those of the NIHR or the Department of Health and Social Care.

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

# A    Proofs in Section 3

## A.1    Proof of Theorem 3.1 based on Restricted Stone-Weierstrass

In this appendix, we provide a proof of the lemmas used in Section 3.1. Before proving them, we report the statement as well.

We start with Lemma 3.1.

**Lemma A.1.** *Let $d \in \mathbb{N}$, $\mathcal{X} \subseteq \mathbb{R}^d$ have at least two points, and $\sigma = \mathrm{ReLU}$. Then $\mathcal{G}_{d,\sigma}(\mathcal{X}, \mathbb{R})$ separates the points of $\mathcal{X}$.*

*Proof.* Consider the functions in $\mathcal{G}_{d,\sigma}(\mathcal{X}, \mathbb{R})$ with $h = 1$ and $L = 0$. These are all and only the 1-Lipschitz affine functions of the form $g(x) = u^\top x + w$, for a pair $u \in \mathbb{R}^d$ and $w \in \mathbb{R}$, with $\|u\|_2 \leq 1$. Let $x, y \in \mathcal{X}$ be two distinct points and $a, b \in \mathbb{R}$ be such that $|a - b| \leq \|y - x\|_2$. Let

$$\widehat{u} = \frac{x - y}{\|x - y\|_2} \in \mathbb{R}^d.$$

Since $\widehat{u}^\top x - \widehat{u}^\top y = \widehat{u}^\top (x - y) = \|x - y\|_2 \neq 0$, the linear system

$$\begin{bmatrix} \widehat{u}^\top x & 1 \\ \widehat{u}^\top y & 1 \end{bmatrix} \begin{bmatrix} \lambda \\ w \end{bmatrix} = \begin{bmatrix} a \\ b \end{bmatrix}$$

has a unique solution. We set $g(z) = \lambda \widehat{u}^\top z + w$, where

$$\lambda = \frac{1}{\|x - y\|_2}(a - b).$$

Since $|a - b| \leq \|x - y\|_2$ we conclude $|\lambda| \leq 1$ and hence if we set $g(x) = u^\top x + w$, with $u = \lambda \widehat{u}$, we get $g \in \mathcal{G}_{d,\sigma}(\mathcal{X}, \mathbb{R})$, $g(x) = a$, and $g(y) = b$ as desired.    $\square$

We now provide the statement and proof of Lemma 3.2.

**Lemma A.2.** *Let $d \in \mathbb{N}$, $\mathcal{X} \subseteq \mathbb{R}^d$, $\sigma = \mathrm{ReLU}$. Consider two functions $f, g \in \mathcal{G}_{d,\sigma}(\mathcal{X}, \mathbb{R})$. There exist $L \in \mathbb{N}$, $h_1, h_2 \in \mathbb{N}$, $v_1 \in \mathbb{R}^{h_1}$, $v_2 \in \mathbb{R}^{h_2}$ with $\|v_1\|_2 = \|v_2\|_2 = 1$, $Q_1 : \mathbb{R}^d \to \mathbb{R}^{h_1}$ and $Q_2 : \mathbb{R}^d \to \mathbb{R}^{h_2}$ affine maps, $\Phi_{\theta_1}, ..., \Phi_{\theta_L} \in \mathcal{E}_{h_1+h_2,\sigma}$, and $M \in \mathbb{R}^{(h_1+h_2) \times (h_1+h_2)}$ symmetric positive semi-definite with $\|M\|_2 \leq 1$, such that*

$$\begin{bmatrix} f(x)v_1 \\ g(x)v_2 \end{bmatrix} = M \circ \Phi_{\theta_L} \circ ... \circ \Phi_{\theta_1} \circ \begin{bmatrix} Q_1 \\ Q_2 \end{bmatrix} x.$$

*Proof.* Consider $f, g \in \mathcal{G}_{d,\sigma}(\mathcal{X}, \mathbb{R})$ that take the form

$$\begin{aligned} f(x) &= v_1^\top \circ \Phi_{\theta_{1,L_1}} \circ ... \circ \Phi_{\theta_{1,1}} \circ Q_1(x) \\ g(x) &= v_2^\top \circ \Phi_{\theta_{2,L_2}} \circ ... \circ \Phi_{\theta_{2,1}} \circ Q_2(x), \end{aligned} \tag{4}$$

with $\Phi_{\theta_{i,\ell}} : \mathbb{R}^{h_i} \to \mathbb{R}^{h_i}$, for $i = 1, 2$ and $\ell = 1, ..., L_i$. Since the identity map belongs to $\mathcal{E}_{h,\sigma}$ for any $h \in \mathbb{N}$, we can assume $L_1 = L_2 = L$. Let $\ell \in \{1, ..., L\}$ and define $\Phi_{\theta_\ell} : \mathbb{R}^{(h_1+h_2)} \to \mathbb{R}^{(h_1+h_2)}$ as $\Phi_{\theta_\ell}(x_1, x_2) = (\Phi_{\theta_{1,\ell}}(x_1), \Phi_{\theta_{2,\ell}}(x_2))$, for every $(x_1, x_2) \in \mathbb{R}^{h_1} \times \mathbb{R}^{h_2}$. More extensively, we can write

$$\Phi_{\theta_\ell}(x_1, x_2) = \begin{pmatrix} x_1 - \tau_{1,\ell}(W_{1,\ell})^\top \sigma(W_{1,\ell}x_1 + b_{1,\ell}) \\ x_2 - \tau_{2,\ell}(W_{2,\ell})^\top \sigma(W_{2,\ell}x_2 + b_{2,\ell}) \end{pmatrix},$$

for a suitable choice of parameters. The dimensions of these parameters are: $\tau_{1,\ell}, \tau_{2,\ell} \in \mathbb{R}$, $W_{1,\ell} \in \mathbb{R}^{h_1,\ell \times h_1}$, $W_{2,\ell} \in \mathbb{R}^{h_2,\ell \times h_2}$, $b_{1,\ell} \in \mathbb{R}^{h_1,\ell}$, and $b_{2,\ell} \in \mathbb{R}^{h_2,\ell}$. Assume, without loss of generality, that $\tau_{1,\ell}, \tau_{2,\ell} \neq 0$ and $\tau_{1,\ell} < \tau_{2,\ell}$. We remark that if either $\tau_{1,\ell}$ or $\tau_{2,\ell}$ were zero, we could get the same map by setting the corresponding weight matrix to zero and replacing the step with any other admissible scalar. Call $\gamma_\ell = \tau_{1,\ell}/\tau_{2,\ell} \in (0, 1)$. It follows that

$$\begin{aligned} &x_1 - \tau_{1,\ell}(W_{1,\ell})^\top \sigma(W_{1,\ell}x_1 + b_{1,\ell}) \\ &= x_1 - \tau_{2,\ell}\gamma_\ell(W_{1,\ell})^\top \sigma(W_{1,\ell}x_1 + b_{1,\ell}) \\ &= x_1 - \tau_{2,\ell}(\sqrt{\gamma_\ell}W_{1,\ell})^\top \sigma(\sqrt{\gamma_\ell}W_{1,\ell}x_1 + \sqrt{\gamma_\ell}b_{1,\ell}) \\ &= x_1 - \tau_{2,\ell}(\widetilde{W}_{1,\ell})^\top \sigma(\widetilde{W}_{1,\ell}x_1 + \widetilde{b}_{1,\ell}), \quad \widetilde{W}_{1,\ell} := \sqrt{\gamma_\ell}W_{1,\ell}, \widetilde{b}_{1,\ell} := \sqrt{\gamma_\ell}b_{1,\ell}, \end{aligned}$$

where we used the positive homogeneity of $\sigma$, i.e., $\sigma(\gamma x) = \gamma\sigma(x)$ for all $x \in \mathbb{R}$ and $\gamma \geq 0$. Since $\sqrt{\gamma} \in (0,1)$, the property $\|\widetilde{W}_\ell\|_2 \leq 1$ is preserved by this manipulation and, therefore, we can assume to have $\tau_{1,\ell} = \tau_{2,\ell} =: \tau_\ell$ for every $\ell = 1, ..., L$. It follows that

$$
\begin{aligned}
\Phi_{\theta_\ell}(x_1, x_2) &= \begin{pmatrix} x_1 - \tau_\ell (W_{1,\ell})^\top \sigma(W_{1,\ell} x_1 + b_{1,\ell}) \\ x_2 - \tau_\ell (W_{2,\ell})^\top \sigma(W_{2,\ell} x_2 + b_{2,\ell}) \end{pmatrix} \\
&= \begin{pmatrix} x_1 \\ x_2 \end{pmatrix} - \tau_\ell \begin{pmatrix} W_{1,\ell} & 0_{h_{1,\ell},h_2} \\ 0_{h_{2,\ell},h_1} & W_{2,\ell} \end{pmatrix}^\top \sigma\left( \begin{pmatrix} W_{1,\ell} & 0_{h_{1,\ell},h_2} \\ 0_{h_{2,\ell},h_1} & W_{2,\ell} \end{pmatrix} \begin{pmatrix} x_1 \\ x_2 \end{pmatrix} + \begin{pmatrix} b_{1,\ell} \\ b_{2,\ell} \end{pmatrix} \right) \\
&= \begin{pmatrix} x_1 \\ x_2 \end{pmatrix} - \tau_\ell \widehat{W}_\ell^\top \sigma\left( \widehat{W}_\ell \begin{pmatrix} x_1 \\ x_2 \end{pmatrix} + \widehat{b}_\ell \right),
\end{aligned}
$$

$$
\widehat{W}_\ell = \begin{pmatrix} W_{1,\ell} & 0_{h_{1,\ell},h_2} \\ 0_{h_{2,\ell},h_1} & W_{2,\ell} \end{pmatrix} \in \mathbb{R}^{(h_{1,\ell}+h_{2,\ell})\times(h_1+h_2)}, \quad \widehat{b}_\ell = \begin{pmatrix} b_{1,\ell} \\ b_{2,\ell} \end{pmatrix} \in \mathbb{R}^{(h_{1,\ell}+h_{2,\ell})}.
$$

We also remark that

$$
\|\widehat{W}_\ell\|_2 = \max\{\|W_{1,\ell}\|_2, \|W_{2,\ell}\|_2\} \leq 1.
$$

Let us then consider the matrix

$$
M = \begin{bmatrix} v_1 v_1^\top & 0_{h_1,h_2} \\ 0_{h_2,h_1} & v_2 v_2^\top \end{bmatrix} \in \mathbb{R}^{(h_1+h_2)\times(h_1+h_2)},
$$

which is symmetric, positive semi-definite and with $\|M\|_2 \leq 1$ as desired. It is immediate to see that $M$ plays the desired role, given the expression for $f$ and $g$ in (4). $\qquad\square$

## A.2 Proof of Theorem 3.1 based on piecewise affine functions

We now provide the statement and proof of Theorem 3.2.

**Theorem A.1.** *Any piecewise affine 1-Lipschitz function $f : \mathbb{R}^d \to \mathbb{R}$ can be represented by a network in $\mathcal{G}_{d,\sigma}(\mathbb{R}^d, \mathbb{R})$ with $\sigma = \mathrm{ReLU}$.*

*Proof.* By Lemma 3.4, there exists a choice of scalars $b_{i,j} \in \mathbb{R}$ and vectors $a_{i,j} \in \mathbb{R}^d$ such that

$$
f(x) = \max\{f_1(x), ..., f_k(x)\}, \ \ f_i(x) = \min\{a_{i,1}^\top x + b_{i,1}, ..., a_{i,l_i}^\top x + b_{i,l_i}\}, \ i = 1, ..., k,
$$

where $\|a_{i,j}\|_2 \leq 1$. Fix $h = l_1 + ... + l_k$. We set $\widehat{Q} \in \mathbb{R}^{h\times d}$ and $\widehat{q} \in \mathbb{R}^h$ as

$$
\widehat{Q} = \begin{bmatrix} a_{1,1}^\top \\ \vdots \\ a_{1,l_1}^\top \\ \vdots \\ a_{k,1}^\top \\ \vdots \\ a_{k,l_k}^\top \end{bmatrix}, \ \ \widehat{q} = \begin{bmatrix} b_{1,1} \\ \vdots \\ b_{1,l_1} \\ \vdots \\ b_{k,1} \\ \vdots \\ b_{k,l_k} \end{bmatrix}.
$$

Then, by Proposition 3.1, we conclude that there exists a choice $\Phi_{\theta_1}, ..., \Phi_{\theta_L}$, with $L \leq (k-1) + (\max\{l_1, ..., l_k\} - 1)$, and a unit-norm vector $v \in \mathbb{R}^h$ such that

$$
f = v^\top \circ \Phi_{\theta_L} \circ ... \circ \Phi_{\theta_1} \circ Q,
$$

where $Q(x) = \widehat{Q}x + \widehat{q}$, as desired. More explicitly, the first $\max\{l_1, ..., l_k\} - 1$ residual layers can be used to assemble the functions $f_1(x), ..., f_k(x)$ in parallel, using block diagonal weight matrices. After these, the remaining $k - 1$ can be used to extract their minimum. We remark that, despite $Q$ is not 1-Lipschitz, the resulting map $f$ is 1-Lipschitz, and hence $f$ belongs to $\mathcal{G}_{d,\sigma}(\mathbb{R}^d, \mathbb{R})$. $\qquad\square$

We now provide the statement and proof of Lemma 3.5 for convex polytopes.

**Lemma A.3.** *The set of piecewise affine 1-Lipschitz functions over $\mathcal{X} \subset \mathbb{R}^d$, a compact and convex polytope, satisfies the universal approximation property for $\mathcal{C}_1(\mathcal{X}, \mathbb{R})$.*

*Proof.* Let $f : \mathcal{X} \to \mathbb{R}$ be an arbitrary 1-Lipschitz function, and fix $\varepsilon > 0$. Consider a covering $\{S_i : i = 1, ..., N\}$ of $\mathcal{X}$ into simplices having intersections that are either trivial, or coincide with a vertex, or with a facet. Assume, without loss of generality, that for every $S_i$, one has

$$\max_{x \in S_i} \min_{x_{i,j} \in \mathcal{V}(S_i)} \|x - x_{i,j}\|_2 \le \varepsilon/2,$$

where $\mathcal{V}(S_i)$ is the set of vertices of $S_i$. Let $g_i : S_i \to \mathbb{R}$ be a $1-$Lipschitz piecewise affine interpolant of $f$ on the vertices of $S_i$, i.e. such that $g_i(x_{i,j}) = f(x_{i,j})$ for every $x_{i,j} \in \mathcal{V}(S_i)$. To build $g_i$, one could for example follow the construction in [25, Proof of Proposition 2.2]. Then, for every $x \in S_i$, there exists $x_{i,j} \in \mathcal{V}(S_i)$ for which

$$|f(x) - g_i(x)| = |f(x) - f(x_{i,j}) + f(x_{i,j}) - g_i(x)| \le 2\|x - x_{i,j}\|_2 \le \varepsilon.$$

We can now glue the local pieces to assemble the piecewise affine continuous function $g(x) = \sum_{i=1}^{N} 1_{S_i}(x) g_i(x)$, where

$$1_{S_i}(x) = \begin{cases} 1, & x \in S_i \\ 0, & x \notin S_i \end{cases}.$$

The Lipschitz constant of $g$ is given by $\text{Lip}(g) = \max_{i=1,...,N} \text{Lip}(g_i) \le 1$. In fact, let $x, y \in \mathcal{X}$ and define the line segment $s(t) = x + t(y-x)$, $t \in [0, 1]$, satisfying $s(0) = x$ and $s(1) = y$. By convexity of $\mathcal{X}$, $s([0, 1]) \subset \mathcal{X}$ holds as well. Thus, there exists a finite set of scalars $0 \le t_1 < t_2 < ... < t_K \le 1$ such that $s(t_i) \in S_{l_i} \cap S_{m_i}$ for a pair of indices $l_i, m_i \in \{1, ..., N\}$. Let us also define $t_0 = 0$ and $t_{K+1} = 1$. We thus have that

$$\sum_{i=0}^{K} \|s(t_{i+1}) - s(t_i)\|_2 = \sum_{i=0}^{K} (t_{i+1} - t_i)\|y - x\|_2 = \|y - x\|_2,$$

and hence

$$|g(y) - g(x)| = \left| \sum_{i=0}^{K} g(s(t_{i+1})) - g(s(t_i)) \right| =_{(\#)} \left| \sum_{i=0}^{K} g_{\ell_i}(s(t_{i+1})) - g_{\ell_i}(s(t_i)) \right|$$

$$\le \sum_{i=0}^{K} \text{Lip}(g_{\ell_i})\|s(t_{i+1}) - s(t_i)\|_2 \le \left( \max_{j=1,...,N} \text{Lip}(g_j) \right) \sum_{i=0}^{K} \|s(t_{i+1}) - s(t_i)\|_2$$

$$= \left( \max_{i=1,...,N} \text{Lip}(g_i) \right) \|y - x\|_2 \le \|y - x\|_2.$$

We remark that the equality in $(\#)$ follows from the convexity of the simplices, which guarantees that the line segment connecting $s(t_i)$ to $s(t_{i+1})$ is fully contained in $S_{l_i}$, and the continuity of $g$ at the boundaries of the simplices.

We conclude that

$$\max_{x \in \mathcal{X}} |f(x) - g(x)| < \varepsilon,$$

where $g$ is piecewise affine and 1-Lipschitz. $\square$

## B    Preliminary results for Section 4

Fix $k \in \mathbb{N}$, and $m \in \mathbb{N}^k$. Call $\alpha_m = \|m\|_1 = m_1 + ... + m_k$. We introduce the set of functions

$$\mathcal{F}_{d,m} = \left\{ F : \mathbb{R}^d \to \mathbb{R}^{\alpha_m} \; \middle| \; F(x) = \begin{bmatrix} f_1(x) \\ f_2(x) \\ \vdots \\ f_k(x) \end{bmatrix}, \; f_i \in \mathcal{C}_1(\mathbb{R}^d, \mathbb{R}^{m_i}), \; i = 1, ..., k \right\}. \tag{5}$$

It is immediate to see that $\mathcal{F}_{d,m} = \mathcal{C}_1(\mathbb{R}^d, \mathbb{R}^m)$ if $m \in \mathbb{N}$, i.e., $k = 1$. Furthermore, for any given $m \in \mathbb{N}^k$, one has $\mathcal{C}_1(\mathbb{R}^d, \mathbb{R}^{\alpha_m}) = \mathcal{F}_{d,\alpha_m} \subset \mathcal{F}_{d,m}$, showing that $\mathcal{F}_{d,m}$ provides a generalisation of the set of 1-Lipschitz functions from $\mathbb{R}^d$ to $\mathbb{R}^{\alpha_m}$. For the results below, we fix a $k \in \mathbb{N}$ and $m \in \mathbb{N}^k$.

**Definition B.1** (Projection on the $i$-th component). *The projection map on the $i$-th component* $\pi_i : \mathbb{R}^{\alpha_m} \to \mathbb{R}^{m_i}$ *is defined as*

$$\mathbb{R}^{\alpha_m} \ni x = \begin{bmatrix} x_1 \\ x_2 \\ \vdots \\ x_k \end{bmatrix} \mapsto x_i =: \pi_i(x), \ x_i \in \mathbb{R}^{m_i}.$$

**Lemma B.1.** *Let $F \in \mathcal{F}_{d,m}$ and $A \in \mathcal{L}_m$. Then, $G = A \circ F : \mathbb{R}^d \to \mathbb{R}^{\alpha_m}$ belongs to $\mathcal{F}_{d,m}$.*

*Proof.* Let $x, y \in \mathbb{R}^d$ be two arbitrary vectors, and fix $i \in \{1, ..., k\}$. By direct calculation, we see that

$$\|\pi_i(G(y) - G(x))\|_2 = \left\| \sum_{j=1}^{k} A_{ij} \pi_j(F(y) - F(x)) \right\|_2 = \left\| \sum_{j=1}^{k} A_{ij}(f_j(y) - f_j(x)) \right\|_2$$

$$\leq \sum_{j=1}^{k} \|A_{ij}\|_2 \|y - x\|_2 \leq \|y - x\|_2$$

as desired. $\qquad\qquad\square$

**Lemma B.2.** *Let $F \in \mathcal{F}_{d,m}$, and $v \in \mathbb{R}^{\alpha_m}$ be a vector with $\|v\|_1 \leq 1$. Then $f(x) = v^\top F(x)$ is a scalar $1-$Lipschitz function.*

*Proof.* Let $x, y \in \mathbb{R}^d$ be two arbitrary vectors. By direct calculation, we see that

$$|f(y) - f(x)| = \left| \sum_{i=1}^{k} \pi_i(v)^\top \pi_i(F(y) - F(x)) \right| \leq \sum_{i=1}^{k} \|\pi_i(v)\|_2 \|\pi_i(F(y) - F(x))\|_2$$

$$\leq \sum_{i=1}^{k} \|\pi_i(v)\|_1 \|y - x\|_2 = \|v\|_1 \|y - x\|_2 \leq \|y - x\|_2$$

as desired. $\qquad\qquad\square$

## C   Proof of Theorem 4.1

We now provide a proof for Lemma 4.1.

**Lemma C.1.** *Let $h \in \mathbb{N}$ and $\sigma = \text{ReLU}$. The set $\widetilde{\mathcal{E}}_{h,\sigma}$ is a subset of $\mathcal{E}_{h+3,\sigma}$.*

*Proof.* Let $x \in \mathbb{R}^{h+3}$, and consider the map

$$\Phi_\theta(x) = \begin{bmatrix} \max\{x_1, x_2\} \\ \min\{x_1, x_2\} \\ x_3 \\ \widetilde{\Phi}_\theta(x_{4:}) \end{bmatrix}$$

with $\widetilde{\Phi}_\theta(x_{4:}) = x_{4:} - \tau \widetilde{W}^\top \sigma(\widetilde{W} x_{4:} + \widetilde{b})$, where $\tau \in [0, 2]$, $\widetilde{W} \in \mathbb{R}^{h' \times h}$, $\widetilde{b} \in \mathbb{R}^{h'}$, and $\|\widetilde{W}\|_2 \leq 1$. Call $\gamma = \sqrt{\frac{\tau}{2}} \in [0, 1]$. Define

$$W = \begin{bmatrix} -1/\sqrt{2} & 1/\sqrt{2} & 0 & 0_h^\top \\ 0 & 0 & 0 & 0_h^\top \\ 0 & 0 & 0 & 0_h^\top \\ 0_{h'} & 0_{h'} & 0_{h'} & \gamma \widetilde{W} \end{bmatrix} \in \mathbb{R}^{(h'+3) \times (h+3)}, \ b = \begin{bmatrix} 0 \\ 0 \\ 0 \\ \gamma \widetilde{b} \end{bmatrix} \in \mathbb{R}^{h'+3}.$$

We now show that $\Phi_\theta(x) = x - 2W^\top\sigma(Wx+b)$:

$$x - 2W^\top\sigma(Wx+b)$$

$$= \begin{bmatrix} x_1 \\ x_2 \\ x_3 \\ x_{4:} \end{bmatrix} - 2\begin{bmatrix} -1/\sqrt{2} & 0 & 0 & 0_{h'}^\top \\ 1/\sqrt{2} & 0 & 0 & 0_{h'}^\top \\ 0 & 0 & 0 & 0_{h'}^\top \\ 0_h & 0_h & 0_h & \gamma\widetilde{W}^\top \end{bmatrix} \sigma\left(\begin{bmatrix} -1/\sqrt{2} & 1/\sqrt{2} & 0 & 0_h^\top \\ 0 & 0 & 0 & 0_h^\top \\ 0 & 0 & 0 & 0_h^\top \\ 0_{h'} & 0_{h'} & 0_{h'} & \gamma\widetilde{W} \end{bmatrix} x + \begin{bmatrix} 0 \\ 0 \\ 0 \\ \gamma\widetilde{b} \end{bmatrix}\right)$$

$$= \begin{bmatrix} x_1 \\ x_2 \\ x_3 \\ x_{4:} \end{bmatrix} - 2\begin{bmatrix} -\frac{1}{2}\sigma(x_2 - x_1) \\ \frac{1}{2}\sigma(x_2 - x_1) \\ 0 \\ \frac{\tau}{2}\widetilde{W}^\top\sigma(\widetilde{W}x_{4:} + \widetilde{b}) \end{bmatrix} = \begin{bmatrix} \max\{x_1, x_2\} \\ \min\{x_1, x_2\} \\ x_3 \\ \Phi_\theta(x_{4:}) \end{bmatrix} = \Phi_\theta(x).$$

$\square$

We now prove Lemma 4.2.

**Lemma C.2.** *Let $\sigma = \mathrm{ReLU}$, $d, h \in \mathbb{N}$, with $h \geq 3$. All the functions in $\widetilde{\mathcal{G}}_{d,\sigma,h}(\mathbb{R}^d, \mathbb{R})$ are 1-Lipschitz.*

*Proof.* Fix $m = (1, 1, 1, h-3)$ and $x \in \mathbb{R}^d$. We consider the network

$$x \mapsto v^\top \circ \Phi_{\theta_L} \circ A_{L-1} \circ ... \circ A_1 \circ \Phi_{\theta_1} \circ Q(x) \in \widetilde{\mathcal{G}}_{d,\sigma,h}(\mathbb{R}^d, \mathbb{R}).$$

For any map $Q \in \mathcal{R}_{d,m}$, there exist four vectors $a_1, a_2, a_3, \tilde{q} \in \mathbb{R}^d$, three scalars $b_1, b_3, b_3 \in \mathbb{R}$, and a matrix $\widetilde{Q} \in \mathbb{R}^{h-3 \times d}$ such that

$$Q(x) = \begin{bmatrix} a_1^\top x + b_1 \\ a_2^\top x + b_2 \\ a_3^\top x + b_3 \\ \widetilde{Q}x + \tilde{q} \end{bmatrix} \in \mathbb{R}^h$$

and $\|a_1\|_2, \|a_2\|_2, \|a_3\|_2, \|\widetilde{Q}\|_2 \leq 1$. Because the composition of 1-Lipschitz maps is 1-Lipschitz, and the maximum and minimum of 1-Lipschitz functions is still 1-Lipschitz, it follows that the map

$$x \mapsto \Phi_{\theta_1} \circ Q(x) = \begin{bmatrix} \max\{a_1^\top x + b_1, a_2^\top x + b_2\} \\ \min\{a_1^\top x + b_1, a_2^\top x + b_2\} \\ a_3^\top x + b_3 \\ \widetilde{Q}x - \tau_1\widetilde{W}_1^\top\sigma(\widetilde{W}_1\widetilde{Q}x + \widetilde{b}_1) \end{bmatrix}$$

belongs to $\mathcal{F}_{d,m}$ defined as in (5). Since $A_1 \in \mathcal{L}_m$, Lemma B.1 implies that

$$A_1 \circ \Phi_{\theta_1} \circ Q(x) = \begin{bmatrix} f_{1,1}(x) \\ f_{1,2}(x) \\ f_{1,3}(x) \\ f_{1,4}(x) \end{bmatrix}, \quad f_{1,1}, f_{1,2}, f_{1,3} : \mathbb{R}^d \to \mathbb{R}, f_{1,4} : \mathbb{R}^d \to \mathbb{R}^{h-3}$$

belongs to $\mathcal{F}_{d,m}$ as well. We leave the components unspecified so that the argument generalises. Doing one further step, so that the argument generalises to a generic number of compositions $L$, one gets

$$\Phi_{\theta_2} \circ A_1 \circ \Phi_{\theta_1} \circ Q(x) = \begin{bmatrix} \max\{f_{1,1}(x), f_{1,2}(x)\} \\ \min\{f_{1,1}(x), f_{1,2}(x)\} \\ f_{1,3}(x) \\ f_{1,4}(x) - \tau_2\widetilde{W}_2^\top\sigma(\widetilde{W}_2 f_{1,4}(x) + \widetilde{b}_2) \end{bmatrix},$$

which again belongs to $\mathcal{F}_{d,m}$. Extending the argument, we see that

$$\Phi_{\theta_L} \circ A_{L-1} \circ ... \circ A_1 \circ \Phi_{\theta_1} \circ Q(x) = \begin{bmatrix} \max\{f_{L-1,1}(x), f_{L-1,2}(x)\} \\ \min\{f_{L-1,1}(x), f_{L-1,2}(x)\} \\ f_{L-1,3}(x) \\ f_{L-1,4}(x) - \tau_L\widetilde{W}_L^\top\sigma(\widetilde{W}_L f_{L-1,4}(x) + \widetilde{b}_L) \end{bmatrix},$$

which belongs to $\mathcal{F}_{d,m}$. Lemma B.2 allows us to conclude that the output scalar function is 1-Lipschitz since $v \in \mathbb{R}^h$ satisfies $\|v\|_1 \leq 1$. $\square$

We conclude with the statement and proof of Theorem 4.2.

**Theorem C.1.** *Any piecewise affine 1-Lipschitz function $f : \mathbb{R}^d \to \mathbb{R}$ can be represented by a network in $\widetilde{\mathcal{G}}_{d,\sigma,h}(\mathbb{R}^d, \mathbb{R})$ with $\sigma = \mathrm{ReLU}$ and $h = d + 3$.*

*Proof.* Fix $m = (1,1,1,d)$. Let us consider a generic 1-Lipschitz piecewise affine function, which, thanks to Lemma 3.4, can be written as

$$f(x) = \max\{f_1(x), ..., f_k(x)\}, \ f_i(x) = \min\{a_{i,1}^\top x + b_{i,1}, ..., a_{i,l_i}^\top x + b_{i,l_i}\}, \ i = 1, ..., k,$$

for a suitable choice of parameters. We also recall that $\|a_{i,j}\|_2 \le 1$ for every $i = 1, ..., k$ and $j = 1, ..., l_i$. Thanks to Proposition 4.1, there exists a suitable choice of weights giving

$$g_1 := \Phi_{\theta_{1,l_1-1}} \circ A_{1,l_1-2} \circ ... \circ A_{1,1} \circ \Phi_{\theta_{1,1}} : \mathbb{R}^h \to \mathbb{R}^h,$$

with $\Phi_{\theta_{1,l_1-1}}, ..., \Phi_{\theta_{1,1}} \in \widetilde{\mathcal{E}}_{d,\sigma}$ and $A_{1,l_1-2}, ..., A_{1,1} \in \mathcal{L}_m$, so that

$$g_1(Q(x)) = \begin{bmatrix} f_1(x) \\ k_{1,2}(x) \\ k_{1,3}(x) \\ x \end{bmatrix} \in \mathbb{R}^{d+3},$$

where we leave the 1-Lipschitz functions $k_{1,2}, k_{1,3} : \mathbb{R}^d \to \mathbb{R}$ unspecified since they do not play a role in our construction. We now introduce a map $\overline{A_1} \in \mathcal{L}_m$ that is characterised by

$$\overline{A_1} \circ g_1 \circ Q(x) = \begin{bmatrix} a_{2,1}^\top x + b_{2,1} \\ a_{2,2}^\top x + b_{2,2} \\ f_1(x) \\ x \end{bmatrix},$$

that is $\overline{A_1} \circ g_1 \circ Q(x) = \widehat{A}_1 g_1(Q(x)) + \widehat{a}_1$ with

$$\widehat{A}_1 = \begin{bmatrix} 0 & 0 & 0 & a_{2,1}^\top \\ 0 & 0 & 0 & a_{2,2}^\top \\ 1 & 0 & 0 & 0_d^\top \\ 0_d & 0_d & 0_d & I_d \end{bmatrix}, \quad \widehat{a}_1 = \begin{bmatrix} b_{2,1} \\ b_{2,2} \\ 0 \\ 0_d \end{bmatrix}.$$

We then define

$$g_2 := \Phi_{\theta_{2,l_2-1}} \circ A_{2,l_2-2} \circ ... \circ A_{2,1} \circ \Phi_{\theta_{2,1}} : \mathbb{R}^h \to \mathbb{R}^h,$$

so that

$$g_2 \circ \overline{A_1} \circ g_1 \circ Q(x) = \begin{bmatrix} f_2(x) \\ k_{2,2}(x) \\ f_1(x) \\ x \end{bmatrix}.$$

Let $S \in \mathcal{L}_m$ be such that

$$S \circ g_2 \circ \overline{A_1} \circ g_1 \circ Q(x) = \begin{bmatrix} f_2(x) \\ f_1(x) \\ k_{2,2}(x) \\ x \end{bmatrix}.$$

We then introduce the residual map $\Psi \in \widetilde{\mathcal{E}}_{d,\sigma}$ having zero weight matrix, i.e., acting over an input $u \in \mathbb{R}^{d+3}$ as

$$\Psi(u) = \begin{bmatrix} \max\{u_1, u_2\} \\ \min\{u_1, u_2\} \\ u_3 \\ u_{4:} \end{bmatrix}.$$

Similarly to $\overline{A_1}$ defined above, we introduce $\overline{A_2} \in \mathcal{L}_m$ so that given a vector $u \in \mathbb{R}^{d+3}$

$$\overline{A_2}(u) = \begin{bmatrix} a_{3,1}^\top u_{4:} + b_{3,1} \\ a_{3,2}^\top u_{4:} + b_{3,2} \\ u_1 \\ u_{4:} \end{bmatrix},$$

and hence

$$\overline{A_2} \circ (\Psi \circ S) \circ \left(g_2 \circ \overline{A_1} \circ g_1 \circ Q(x)\right) = \begin{bmatrix} a_{3,1}^\top x + b_{3,1} \\ a_{3,2}^\top x + b_{3,2} \\ \max\{f_1(x), f_2(x)\} \\ x \end{bmatrix}.$$

Iterating this reasoning, one can get

$$e_3^\top \circ \overline{A_k} \circ (\Psi \circ S) \circ g_k \circ ... \circ \overline{A_3} \circ (\Psi \circ S) \circ g_3 \circ \overline{A_2} \circ (\Psi \circ S) \circ g_2 \circ \overline{A_1} \circ g_1 \circ Q(x) = f(x),$$

where $e_3 \in \mathbb{R}^{d+3}$ is the third vector of the canonical basis. $\qquad\square$

# D   Extension of the set $\widetilde{\mathcal{E}}_{h,\sigma}$

We now fix $\sigma = \mathrm{ReLU}$. The set of networks $\widetilde{\mathcal{G}}_{d,\sigma,h}(\mathcal{X}, \mathbb{R})$ that we considered, is a subset of

$$\overline{\mathcal{G}}_{d,\sigma,h}(\mathcal{X}, \mathbb{R}) := \Big\{ v^\top \circ \Phi_{\theta_L} \circ A_{L-1} \circ \cdots \circ \Phi_{\theta_2} \circ A_1 \circ \Phi_{\theta_1} \circ Q : \mathcal{X} \to \mathbb{R} \ \Big| \ m = (1, ..., 1, h-k),$$

$$h = \|m\|_1, \ Q \in \mathcal{R}_{d,m}, v \in \mathbb{R}^h, \|v\|_1 \le 1, A_1, ..., A_{L-1} \in \mathcal{L}_m, \Phi_{\theta_\ell} \in \overline{\mathcal{E}}_{h-k,k,\sigma}, \ L \in \mathbb{N}, \ k \in \{1, ..., h\} \Big\},$$

where we define

$$\overline{\mathcal{E}}_{h,k,\sigma} = \Big\{ \Phi_\theta : \mathbb{R}^{h+k} \to \mathbb{R}^{h+k} \ \Big| \ \Phi_\theta(x) = \begin{bmatrix} \mathrm{GroupSort}(x_{1:k}; g) \\ \widetilde{\Phi}_\theta(x_{(k+1):}) \end{bmatrix},$$

$$\widetilde{\Phi}_\theta \in \mathcal{E}_{h,\sigma}, \ g \in \{1, ..., k\} \Big\}.$$

The $\mathrm{GroupSort}(\cdot; g)$ activation function, introduced in [1], splits the input into groups of a specific size $g$ and sorts each in descending order. The particular case of group size $g = 2$ coincides with the MaxMin function we used to define $\widetilde{\mathcal{G}}_{d,\sigma,h}(\mathcal{X}, \mathbb{R})$, i.e.,

$$\mathrm{GroupSort}(x_{1:3}; 2) = \begin{bmatrix} \max\{x_1, x_2\} \\ \min\{x_1, x_2\} \\ x_3 \end{bmatrix}.$$

We can thus say that $\widetilde{\mathcal{E}}_{h,\sigma} \subset \overline{\mathcal{E}}_{h,3,\sigma}$, and hence $\widetilde{\mathcal{G}}_{d,\sigma,h}(\mathcal{X}, \mathbb{R}) \subset \overline{\mathcal{G}}_{d,\sigma,h}(\mathcal{X}, \mathbb{R})$.

The $\mathrm{GroupSort}(\cdot; g)$ activation function might not have symmetric Jacobian, and hence might not be directly expressible with a single negative gradient step. Still, it can be written as a composition of these residual maps in $\mathcal{E}_{h,\sigma}$. This is an immediate consequence of the fact that one can sort a list of numbers by iteratively ordering subgroups of size two. For example, assume there is a positive-measure subregion $\mathcal{R}$ of $\mathcal{X}$ where $x_2 < x_1 < x_3$ for every $x \in \mathcal{R}$. There, the function $\mathrm{GroupSort}(\cdot, 3)$ would act as

$$\mathrm{GroupSort}(x_{1:3}) = \begin{bmatrix} x_3 \\ x_1 \\ x_2 \end{bmatrix} = \begin{bmatrix} 0 & 0 & 1 \\ 1 & 0 & 0 \\ 0 & 1 & 0 \end{bmatrix} x_{1:3} =: P x_{1:3}.$$

The permutation matrix $P$ is not symmetric, and hence the target map can not be expressed with a single negative gradient Euler step. Still, we can write it as the following composition

$$\begin{bmatrix} x_1 \\ x_2 \\ x_3 \end{bmatrix} \mapsto \begin{bmatrix} x_2 \\ x_1 \\ x_3 \end{bmatrix} \mapsto \begin{bmatrix} x_3 \\ x_1 \\ x_2 \end{bmatrix}$$

which are two maps that belong to $\mathcal{E}_{3,\sigma}$. More generally, we can represent the map $\mathrm{GroupSort}(\cdot; 3)$ as follows

$$\begin{bmatrix} x_1 \\ x_2 \\ x_3 \end{bmatrix} \mapsto \begin{bmatrix} \max\{x_1, x_2\} \\ \min\{x_1, x_2\} \\ x_3 \end{bmatrix} \mapsto \begin{bmatrix} \max\{x_1, x_2, x_3\} \\ \min\{x_1, x_2\} \\ \min\{x_3, \max\{x_1, x_2\}\} \end{bmatrix}$$

$$\mapsto \begin{bmatrix} \max\{x_1, x_2, x_3\} \\ \max\{\min\{x_1, x_2\}, \min\{x_3, \max\{x_1, x_2\}\}\} \\ \min\{\min\{x_1, x_2\}, \min\{x_3, \max\{x_1, x_2\}\}\} \end{bmatrix}$$

$$= \begin{bmatrix} \max\{x_1, x_2, x_3\} \\ \max\{\min\{x_1, x_2\}, \min\{x_3, \max\{x_1, x_2\}\}\} \\ \min\{x_1, x_2, x_3\} \end{bmatrix} = \mathrm{GroupSort}(x_{1:3}; 3),$$

which are all maps that belong to $\mathcal{E}_{3,\sigma}$.

$\text{GroupSort}(\cdot, g)$ provides an output with the same scalar components as the input, just reordered. Thus, if applied to a function having all the components which are 1-Lipschitz, the same property is preserved after its application. Combining these results, together with the derivations done for $\widetilde{\mathcal{G}}_{d,\sigma,h}(\mathcal{X}, \mathbb{R})$, it is easy to derive the following results.

**Lemma D.1.** *Let $d \in \mathbb{N}$, $\mathcal{X} \subseteq \mathbb{R}^d$, and $\sigma = \text{ReLU}$. All the functions in $\overline{\mathcal{G}}_{d,\sigma,h}(\mathcal{X}, \mathbb{R})$ are 1-Lipschitz.*

**Lemma D.2.** *Let $d \in \mathbb{N}$, $\mathcal{X} \subset \mathbb{R}^d$ a compact set, and $\sigma = \text{ReLU}$. The set $\overline{\mathcal{G}}_{d,\sigma,h}(\mathcal{X}, \mathbb{R})$ satisfies the universal approximation property for $\mathcal{C}_1(\mathcal{X}, \mathbb{R})$ if $h = d + 3$.*

# E  Extension to multivalued functions

We now state and prove Lemma 5.1.

**Lemma E.1.** *Let $c, d \in \mathbb{N}$, $\mathcal{X} \subset \mathbb{R}^d$ be compact, and $\sigma = \text{ReLU}$. Define the set*

$$\mathcal{G}_{c,d,\sigma}(\mathcal{X}, \mathbb{R}^c) := \Big\{ P \circ \Phi_{\theta_L} \circ \cdots \circ \Phi_{\theta_1} \circ Q : \mathcal{X} \to \mathbb{R}^c \ \Big| \ Q(x) = \widehat{Q}x + \widehat{q}, \ \widehat{Q} \in \mathbb{R}^{h \times d},$$
$$\widehat{q} \in \mathbb{R}^h, \ P \in \mathbb{R}^{c \times h}, \ \|P\|_{2,\infty} = 1, \ \Phi_{\theta_\ell} \in \mathcal{E}_{h,\sigma}, \ L, h \in \mathbb{N} \Big\}$$

*Then, for any $f \in \mathcal{C}_1(\mathcal{X}, \mathbb{R}^c)$ and any $\varepsilon > 0$, there exists $g \in \mathcal{G}_{c,d,\sigma}(\mathcal{X}, \mathbb{R}^c)$ such that $\max_{x \in \mathcal{X}} \|f(x) - g(x)\|_2 \leq \varepsilon$.*

*Proof.* Let $f \in \mathcal{C}_1(\mathcal{X}, \mathbb{R}^c)$, and $\varepsilon > 0$. Call $f_i : \mathcal{X} \to \mathbb{R}$, $i = 1, ..., c$, the $i$-th component of $f$, which belongs to $\mathcal{C}_1(\mathcal{X}, \mathbb{R})$. Since $\mathcal{G}_{d,\sigma}(\mathcal{X}, \mathbb{R}) \subset \mathcal{G}_{1,d,\sigma}(\mathcal{X}, \mathbb{R}^1)$ satisfies the universal approximation property for $\mathcal{C}_1(\mathcal{X}, \mathbb{R})$, for every $i = 1, ..., c$, there exist $g_1, ..., g_c \in \mathcal{G}_{1,d,\sigma}(\mathcal{X}, \mathbb{R}^1)$ such that

$$\max_{x \in \mathcal{X}} |f_i(x) - g_i(x)| \leq \frac{\varepsilon}{\sqrt{c}}, \ i = 1, ..., c.$$

The residual maps can represent the identity, and we can thus assume that the number of layers of $g_1, ..., g_c$ coincide and are equal to $L$. Call $h_i \in \mathbb{N}$, $Q_i : \mathbb{R}^d \to \mathbb{R}^{h_i}$, $\Phi_{\theta_{i,\ell}} \in \mathcal{E}_{h_i,\sigma}$ with $\ell = 1, ..., L$, and $v_i \in \mathbb{R}^{h_i}$, the widths, affine lifting layers, residual layers, and linear projection layers of the $c$ scalar-valued networks, respectively. Following similar arguments as for the proof of Theorem 3.1, it is easy to see that

$$x \mapsto g(x) := \begin{bmatrix} g_1(x) \\ \vdots \\ g_c(x) \end{bmatrix}$$

belongs to $\mathcal{G}_{c,d,\sigma}(\mathcal{X}, \mathbb{R}^c)$. Call $h = h_1 + ... + h_c$, and let $Q : \mathbb{R}^d \to \mathbb{R}^h$ be characterised as

$$Q(x) = \begin{bmatrix} Q_1(x) \\ \vdots \\ Q_c(x) \end{bmatrix} \in \mathbb{R}^h.$$

Denote with $\Phi_{\theta_\ell} \in \mathcal{E}_{h,\sigma}$, $\ell = 1, ..., L$, the maps

$$\Phi_{\theta_\ell}(u) = \begin{bmatrix} \Phi_{\theta_{1,\ell}}(u_{1:h_1}) \\ \Phi_{\theta_{2,\ell}}(u_{h_1+1:h_1+h_2}) \\ \vdots \\ \Phi_{\theta_{c,\ell}}(u_{h-h_c+1:h}) \end{bmatrix},$$

where $u_{i:j} \in \mathbb{R}^{j-i+1}$ denotes the entries from position $i$ to position $j$ of $u \in \mathbb{R}^h$. Finally, denote with $P \in \mathbb{R}^{c \times h}$ the projection matrix defined as

$$P = \begin{bmatrix} v_1^\top & 0_{h_2}^\top & \cdots & 0_{h_c}^\top \\ 0_{h_1}^\top & v_2^\top & \cdots & 0_{h_c}^\top \\ \vdots & \ddots & \ddots & \vdots \\ 0_{h_1}^\top & 0_{h_2}^\top & \cdots & v_c^\top \end{bmatrix}.$$

Since $\|v_i\|_2 = 1$ for every $i = 1, ..., c$, we have $\|P\|_{2,\infty} = \max_{i=1,...,c} \|e_i^\top P\|_2 = \max_{i=1,...,c} \|v_i\|_2 = 1$. Therefore, since

$$g(x) = P \circ \Phi_{\theta_L} \circ ... \circ \Phi_{\theta_1} \circ Q(x),$$

we conclude that $g \in \mathcal{G}_{c,d,\sigma}(\mathcal{X}, \mathbb{R}^c)$ as desired. Furthermore, we see that for any given $x \in \mathcal{X}$ one has

$$\|f(x) - g(x)\|_2 = \sqrt{\sum_{i=1}^c |f_i(x) - g_i(x)|^2} \leq \sqrt{c\varepsilon^2/c} = \varepsilon$$

as desired. Thus, for any $f \in \mathcal{C}_1(\mathcal{X}, \mathbb{R}^c)$ and any $\varepsilon > 0$, there exists $g \in \mathcal{G}_{c,d,\sigma}(\mathcal{X}, \mathbb{R}^c)$ such that

$$\max_{x \in \mathcal{X}} \|f(x) - g(x)\|_2 \leq \varepsilon.$$

$\square$

## F Additional comments

### F.1 Further motivation for $1$-Lipschitz ResNets

We provide further motivation for studying 1-Lipschitz ResNets.

#### F.1.1 Why study $1$-Lipschitz networks?

Regarding the importance of 1-Lipschitz networks, we have already cited some works relying on them in the related work section. We expand on how essential the 1-Lipschitz constraint is in those contexts. To do so, we also more explicitly describe the following two problems where 1-Lipschitz constraints are desirable:

1. Improving the network robustness to adversarial attacks, see for example [12, 29, 33];

2. Deploying a network trained as an image denoiser in a Plug-and-Play algorithm for image reconstruction in inverse problems, for example, for deblurring problems, see [6, 16, 33].

In general, in all those situations where one is interested in learning maps that are then iteratively applied, aiming to converge to a fixed point, it is possible to guarantee the convergence of the iterative scheme only under some Lipschitz constraints. Similarly, in those problems where one needs to parametrise the lattice of 1-Lipschitz functions, such as to compute the Wasserstein 1-distance, it is also convenient to rely on these constrained architectures.

**Improving the robustness to adversarial attacks** Training neural networks to solve classification problems is a very standard methodology. Still, it is often the case that networks trained to be accurate classifiers, are very sensitive to perturbations in the inputs. Some of these perturbations, called adversarial, can be constructed with the sole purpose of fooling the network into misclassifying them, despite being quantifiably close to data points drawn from the same probability distribution as the training and test sets. A way to improve the network's robustness to these kinds of perturbations is by controlling its Lipschitz constant. This approach was considered in several papers before ours. A theoretical justification for this methodology can be found by relating the notion of margin to that of Lipschitz regularity. Consider a network $\mathcal{N}_\theta : \mathbb{R}^d \to \mathbb{R}^c$ returning probability vectors in $\mathbb{R}^c$, where $c$ is the number of classes in which we want to partition our dataset. The classification margin at a point $x$ is defined as

$$\mathcal{M}_{\mathcal{N}_\theta}(x) := \mathcal{N}_\theta(x)^\top e_{\ell(x)} - \max_{\substack{j \neq \ell(x) \\ j \in \{1,...,c\}}} \mathcal{N}_\theta(x)^\top e_j,$$

where $\ell(x)$ represents the index of the class to which $x$ belongs. If the margin is positive, then the network correctly classifies $x$. Furthermore, the higher the margin, the more certain the network is of this prediction. One can also prove, see [36], that

$$\mathcal{M}_{\mathcal{N}_\theta}(x) > \sqrt{2}\mathrm{Lip}(\mathcal{N}_\theta)\varepsilon \implies \mathcal{M}_{\mathcal{N}_\theta}(x + \eta) > 0, \ \forall \eta \in \mathbb{R}^d, \|\eta\|_2 \leq \varepsilon.$$

This condition means that if we have a good trade-off between classification margin and Lipschitz constant of the network, we can guarantee the correct classification of perturbed inputs. Such an analysis implies that if we force the network to have a small Lipschitz constant, such as $\mathrm{Lip}(\mathcal{N}_\theta) \leq 1$ as in our paper, and train the network with a loss function aiming to maximise the margin such as the multi-class classification hinge loss, we can get improved robustness.

**Plug-and-Play algorithm with guaranteed convergence properties**   The Banach fixed-point Theorem guarantees that for a map $T : \mathbb{R}^d \to \mathbb{R}^d$ which is strictly 1-Lipschitz, i.e., $\|T(y) - T(x)\|_2 < \|y - x\|_2$ for every $x, y \in \mathbb{R}^d$, there exists a unique fixed point and the iteration $x_{k+1} = T(x_k)$ will converge to it whatever $x_0 \in \mathbb{R}^d$ is. It is thus intuitive that once we are interested in (partially) modelling iterative schemes through neural networks, it is fundamental to rely on architectures constrained as in our paper to get a reliable method. This is the idea behind the Plug-and-Play algorithm [16, 32, 33] used in inverse problems. This method is inspired by the forward-backwards splitting proximal gradient descent approach used to solve

$$\min_{x \in \mathbb{R}^d} f(x) + g(x), \ f : \mathbb{R}^d \to \mathbb{R}, \ g : \mathbb{R}^d \to \mathbb{R} \cup \{+\infty\}.$$

It is common to have $f$ representing the data-fidelity term in the inverse problem, such as $f(x) = \|Kx - y\|_2^2/2$, whereas $g$ represents a regularisation term, such as $g(x) = \|x\|_1$. Typically, one can not ask for $g$ to be continuously differentiable, and hence a methodology to efficiently solve this minimisation problem is the proximal algorithm

$$x_{k+1} = \mathrm{prox}_{g,\tau} \left(x_k - \tau \nabla f(x_k)\right), \ \mathrm{prox}_{g,\tau}(x) = \arg\min_{y \in \mathbb{R}^d} \left( \frac{1}{2\tau} \|y - x\|_2^2 + g(y) \right). \tag{6}$$

There are two main limitations with this procedure for a general inverse problem:

1. It is challenging to design a good regulariser $g$,
2. The proximal operator $\mathrm{prox}_{g,\tau}$ of a generic regulariser $g$ does not admit a closed form.

The Plug-and-Play algorithm addresses both limitations above by replacing $\mathrm{prox}_{g,\tau}$ with a neural network $\mathcal{N}_\theta : \mathbb{R}^d \to \mathbb{R}^d$ trained to denoise inputs. In order to guarantee the convergence of this hybrid scheme

$$x_{k+1} = \mathcal{N}_\theta(x_k - \tau \nabla f(x_k)), \tag{7}$$

we need to properly constrain $\mathcal{N}_\theta$. If $f$ is strongly convex, $L$-smooth, and continuously differentiable, then, taken $\tau \in (0, 2/L)$, the method in (7) converges to a fixed point whenever $\mathcal{N}_\theta$ is 1-Lipschitz. This is another reason why investigating the architecture studied in this paper is fundamental. Weaker convergence guarantees can also be obtained when $f$ is only convex, but more needs to be asked from $\mathcal{N}_\theta$, see, e.g., [33].

**Why study ResNets?**   There are three main reasons why we focus on ResNets:

1. 1-Lipschitz constrained ResNets have been used extensively in several applications;
2. A theory similar to ours for feedforward networks has already been developed;
3. GNNs, Transformers, and other architectures also rely on residual connections which are hence an essential piece to understand.

More explicitly, the gradient steps studied in this paper have already been considered to design GNNs (see, for example, [5, 11, 12]). To the best of our knowledge, these gradient steps have not been used in Transformers. Still, there has already been interest in studying Lipschitz-continuous transformers (see [3, 18, 30]), and our theoretical analysis could at least partially be applied to these more complex architectures.

**Theoretical motivation.**   While unconstrained ResNets can approximate an arbitrary continuous function, it is not necessarily the case that the same holds for a constrained one. For example, in [25, Proposition 3.3], the authors prove that feedforward networks with unit-norm weights, which are 1-Lipschitz, are not dense in the set of 1-Lipschitz functions. This holds despite the unconstrained models being universal approximators of continuous functions. This highlights the necessity of developing an approximation theory which is specific to 1-Lipschitz architectures such as ResNets.

## F.2 Dependence of the network size on the input dimension

We now comment on how the network width and depth grow with the input dimension based on our theoretical analysis.

**Depth.** In Appendix A.2 we derive that to represent the $\max$ and $\min$ functions of affine pieces, we need a network whose depth grows linearly with the number of pieces. This is the primary operation required to understand how the network depth evolves based on our theory. Still, there are two essential comments to complement this discussion:

1. First, our proof is designed to be easy to follow and does not aim to present the most efficient network solving the task. It is in fact evident that a tighter bound for the networks with unbounded width could be to have $L$ belonging to $\mathcal{O}(\log_2(d))$, since $\max\{x_1, ..., x_d\}$ can be written by computing in parallel $d/2$ pairwise maxima and iterating the process, leading to $k$ steps where $k \in \mathbb{N}$ satisfies $(d/2^k) \approx 1$, i.e. $k \approx \log_2(d)$. This does not immediately translate to the finite width case, but one could increase the fixed width and improve the efficiency in terms of network depth.

2. Second, our proof is constructive and provides an upper bound on the size of the network one needs. When training these models, more efficient weight choices could be made, and a more effective growth factor may emerge.

**Width.** For the networks in Theorem 3.1, the network width grows with the number of affine pieces needed to assemble the function. This is function-dependent, and it can not be bounded a priori solely based on the input dimension. Still, we also provide an approximation theorem with fixed width, which does not suffer from this unbounded growth of the width.

**Approximation rates.** One of our proof strategies involves the representation of an arbitrary 1-Lipschitz piecewise affine function with the considered networks. We thus inherit the same approximation rates as this class of functions.

## F.3 Further discussion of other activation functions

Our theoretical analysis focuses on the $\mathrm{ReLU}$ activation function and the techniques we work with heavily rely on the properties of such a function. The focus on a specific activation, and in particular on $\mathrm{ReLU}$, is quite common in the mathematics of deep learning, as can be seen, for example, in [15, 26, 41].

For any other continuous activation $\sigma : \mathbb{R} \to \mathbb{R}$ which is not a polynomial, it is relatively immediate to recover the density of the set of networks

$$\mathcal{G}_d^\sigma(\mathcal{X}, \mathbb{R}) := \{\mathcal{N}_\theta \in \mathcal{G}_{d,f}(\mathcal{X}, \mathbb{R}) : \; f : \mathbb{R} \to \mathbb{R}, \; f(s) = u_1^\top \sigma(u_2 s + u_3), \; u_1, u_2, u_3 \in \mathbb{R}^h, \; h \in \mathbb{N}\}$$

in $\mathcal{C}_1(\mathcal{X}, \mathbb{R})$. To prove this, one can approximate the elements in $\mathcal{G}_{d,\mathrm{ReLU}}(\mathcal{X}, \mathbb{R})$ to an arbitrary accuracy with elements in $\mathcal{G}_d^\sigma(\mathcal{X}, \mathbb{R})$. Such an approximation can be done because the considered residual layers are gradient steps for potential energies of the form $g_\ell(x) = 1_{h_\ell}^\top \mathrm{ReLU}^2(W_\ell x + b_\ell)/2$ and, on compact sets, we can approximate $\mathrm{ReLU}^2/2$ and $\mathrm{ReLU}$ with a single hidden layer network and its derivative, see [28, Theorem 4.1]. Such an analysis would allow us to recover an approximation theory for networks whose residual layers are of the form

$$x \mapsto x - \tau_\ell W_\ell^\top f(W_\ell x + b_\ell).$$

In this case, though, the imposition of the 1-Lipschitz constraints in practice becomes even more challenging, given the lack of knowledge of the properties of $f$, e.g., if it is non-decreasing. A particularly simple example is $\sigma(x) = \mathrm{LeakyReLU}_\alpha(x) = \max\{\alpha x, x\}$, $\alpha \in (0, 1)$ where we do not need approximations since

$$\mathrm{ReLU}(x) = \frac{1}{1 - \alpha^2} \left(\mathrm{LeakyReLU}_\alpha(x) + \alpha \mathrm{LeakyReLU}_\alpha(-x)\right).$$

# G   Numerical experiments and implementability details

This paper focuses on the theoretical analysis of 1-Lipschitz ResNets. We have commented in Section 5 on the implementability of the considered models, and we now demonstrate that they do not have any intrinsic issues with trainability.

To do so, we consider two classification problems. First, we train models to classify the two-moon dataset with additive Gaussian noise of standard deviation $\sigma = 0.1$. Then, we train networks to classify the MNIST dataset. We inspect how the performance varies as we consider models coming from the two families of architectures presented in this paper, and as we vary the network depth and width.

For the two-moon dataset, we generate 4,000 points, $20\%$ of which are set as training points. For MNIST, we adopt the standard training/test split and preprocess the inputs by normalising them. The batch size we consider for both tasks is 256. We optimise the weights for both tasks using Adam and a cosine annealing learning rate scheduler. The experiments are run on a Quadro RTX 6000 GPU.

The network weights can be initialised to satisfy the dynamical isometry property [22, 39], to avoid possible trainability problems when considering deep networks. More explicitly, one can set the intermediate affine layers $A_\ell$ so they realise an isometry, the time steps $\tau_\ell = 2$, and the weights $W_\ell$ in the residual layers to random orthogonal matrices. Such an initialisation strategy allows us to have layers which, at initialisation, are maps with a Jacobian matrix which is orthogonal almost everywhere. The dynamical isometry theory suggests that initialising the network weights so the input-to-output Jacobian is orthogonal almost everywhere is sufficient to train deep neural networks. The reason why the setup written above allows to get the Jacobian orthogonality of the residual layers is that

$$\phi_{\theta_\ell}(x)' = I_h - 2W_\ell^\top D(x)W_\ell, \ D(x) = \text{diag}(\text{ReLU}'(W_\ell x + w_\ell))$$
$$(\phi_{\theta_\ell}'(x))^\top \phi_{\theta_\ell}'(x) = I_h + 4(W_\ell^\top D^2(x)W_\ell - W_\ell^\top D(x)W_\ell) = I_h,$$

and the same holds for $(\phi_{\theta_\ell}'(x)')^\top \phi_{\theta_\ell}'(x) = I_h$, since $D(x)^2 = D(x)$ because of the zero and one slopes of ReLU.

All the layers are constrained as discussed in the main body of the paper, but the first one is left unconstrained to have fair comparisons between the two types of models, i.e., those stemming from Theorem 3.1 and Theorem 4.1. We consider models with a varying number of layers and present the test accuracy as this number changes, along with the time required to enforce the constraints on the network weights during training, computed as an average over the number of epochs using the `time` library in Python. The weight constraints are enforced in a projected gradient descent fashion, suitably normalising the weights after a gradient step. The code is written in PyTorch and it is available at the repository `https://github.com/davidemurari/1LipschitzResNets`.

To compute the spectral norms of the weights and enforce the necessary constraints, we use the power method as described in [24, 33]. To ensure an efficient implementation, we perform several power iterations at initialisation to obtain an accurate approximation of the leading singular vector. We then perform only one iteration per gradient step, as the weights tend to change very mildly during training, and these pre-computed singular vectors provide accurate initial guesses. The cost of this normalisation step is negligible compared with the overall cost of the forward and backwards passes.

The models deriving from Theorem 3.1 and Theorem 4.1 consistently train regardless of the network width and depth. The cost of constraining them is larger for those with affine layers, and it grows faster as the depth increases than when the width does. Still, the implementation could be optimised, and this step could be made faster. In Tables 1 and 2, we denote with Theorem 3.1 the architecture without affine layers, and with Theorem 4.1, with affine layers and further constraints on the gradient steps.

The experiments for MNIST in Table 3 are performed only for the network newly proposed in Theorem 4.1. This is because for the one in Theorem 3.1 there have already been more extensive experimental studies in the past, e.g., [23, 29, 33]. We remark that this task does not entirely represent our theoretical results, since we focus on approximating scalar-valued functions. In contrast, in this case, we have outputs in $\mathbb{R}^{10}$, given that 10 is the number of classes in the MNIST dataset. For this reason, we do not constrain the last affine layer either. The value of these experiments lies in

| Number layers | Theorem 3.1 | | Theorem 4.1 | |
|---|---|---|---|---|
| | **Test Accuracy** | **Normalisation Time** (seconds) | **Test Accuracy** | **Normalisation Time** (seconds) |
| $L = 2$ | 99.75% | 0.0079 | 88.25% | 0.053 |
| $L = 4$ | 99.88% | 0.0130 | 99.88% | 0.097 |
| $L = 8$ | 100.00% | 0.0286 | 99.88% | 0.185 |
| $L = 16$ | 100.00% | 0.0585 | 100.00% | 0.273 |
| $L = 32$ | 99.88% | 0.1091 | 100.00% | 0.326 |
| $L = 64$ | 100.00% | 0.1628 | 100.00% | 0.685 |

Table 1: Networks with varying depth and fixed width. Numerical experiments for the two–moon dataset with additive Gaussian noise of standard deviation $\sigma = 0.1$. The hidden dimension is set to $d + 3 = 5$, where $d = 2$ is the input dimension of the data points.

| Network width | Theorem 3.1 | | Theorem 4.1 | |
|---|---|---|---|---|
| | **Test Accuracy** | **Normalisation Time** (seconds) | **Test Accuracy** | **Normalisation Time** (seconds) |
| $h = 10$ | 99.88% | 0.0372 | 100.00% | 0.1628 |
| $h = 20$ | 99.88% | 0.0379 | 99.88% | 0.1744 |
| $h = 30$ | 99.88% | 0.0381 | 99.88% | 0.1829 |
| $h = 40$ | 100.00% | 0.0378 | 99.63% | 0.2993 |

Table 2: Networks with fixed depth and varying width. Numerical experiments for the two–moon dataset with additive Gaussian noise of standard deviation $\sigma = 0.1$. The number of layers is fixed to $L = 10$ while the hidden dimension $h$ varies.

strengthening the claims about the trainability of these newly proposed models, which, despite the addition of interleaved affine layers, do not suffer from practical problems.

| | $L = 5$ | $L = 10$ | $L = 20$ |
|---|---|---|---|
| $h = 50$ | 97.85% | 97.67% | 97.82% |
| $h = 100$ | 97.94% | 97.70% | 97.58% |
| $h = 200$ | 97.68% | 97.77% | 97.89% |

Table 3: Results on MNIST with the network in Theorem 4.1: test accuracy for different widths $h$ and depths $L$.

