# OpenReview forum: "Approximation theory for 1-Lipschitz ResNets"
_NeurIPS.cc/2025/Conference — NeurIPS 2025 poster_

### Official Review · Reviewer_KS54 · 2025-06-13

**Clarity:** 3
**Significance:** 3
**Originality:** 3
**Rating:** 5
**Confidence:** 4

**Summary:**

In this paper, the authors study the approximation properties of 1-Lipschitz ResNets. The main results show that the class of 1-Lipschitz ResNests is the universal approximator of the 1-Lipschitz functions.

**Questions:**

Please see the above comments.

**Ethical Concerns:**

["NO or VERY MINOR ethics concerns only"]

**Final Justification:**

After carefully reading the rebuttal again, I think all the concerns have been properly addressed. However, I think it would add value to the paper if the discussion about the dependence of approximation error on dimension is added in the main paper. I have increased my score.

**Limitations:**

Please see the above comments.

**Quality:**

3

**Strengths And Weaknesses:**

# Strength
The paper is describes the problem clearly and the claims are easy to understand. The results presented in the paper seems to be useful as it theoretically explains the usefulness of ResNets. In my opinion the paper answers an important and interesting open problem in this area.

# Weakness

Although the main results are strong, I pose a couple of important questions below:

- Both the main results do not mention the number of Layers $L$ needed to get a desired level of approximations. Typically, in applications, we have a very high data dimension $d$. If $L$ depends linearly on $d$, then it would be very hard to train the model and would seriously hinder the implementability.

- Is it possible to quantify the approximation error in terms of the data dimension $d$? This has a practical importance as good approximation properties will help to implelemnt comparatively less complex models that are easier to train.

---

> ### Author Rebuttal · Authors · 2025-07-29
>
> Thank you for pointing out these two questions, which we agree are essential aspects to discuss. We expand on them below.
>
> ## Weaknesses
> ### How does the number of layers grow?
>
> We already specify an upper bound on the number of layers needed to assemble the $\max$ and $\min$ functions of the affine pieces, which grows at most linearly. This can be found, for example, in line 454 of our manuscript. There are two essential aspects to address your concern.
> 1. First, our proof is designed to be easy to follow and does not aim to present the most efficient network solving the task. A tighter bound for the networks with unbounded width could be to have $L$ in $\mathcal{O}(\log_2(d))$, since we can write $\max\\{x_1,...,x_d\\}$ by computing in parallel $d/2$ pairwise maxima and keep iterating the process, leading to $k$ steps where $k\in\mathbb{N}$ satisfies $(d/2^k)\approx 1$, i.e. $k\approx \log_2(d)$. This does not immediately translate to the finite width case, but one could increase the fixed width and improve the efficiency in terms of network depth.
> 2. Second, our proof is constructive and provides an upper bound on the size of the network one needs. When training these models, more efficient weight choices could be made, and a more effective growth factor may emerge. We now discuss these aspects further in the paper.
>
> ### How does the approximation error depend on the input dimension?
>
> Based on our proof strategy involving the representation of an arbitrary $1$-Lipschitz piecewise affine function, we inherit the same approximation rates as this class of functions. To get such a rate, we would need to rely on theoretical results of the following form: Let $\Omega\subset\mathbb{R}^d$ be a compact set. For any $1$-Lipschitz function $f:\Omega\to\mathbb{R}$, and each $\varepsilon>0$ there is a piecewise-affine $1$-Lipschitz function with $\mathcal{O}(d^k/\varepsilon^m)$, $k,m\in\mathbb{Q}$, affine pieces approximating $f$ within tolerance $\varepsilon$. We are currently not aware of results of this kind, but they would be sufficient to explore such rates for the networks we study. Thank you for bringing this interesting theoretical question to our attention, which we will explore further in the future.
>
> Thank you for the careful analysis of our manuscript, and we hope that these responses address your questions.

---

> > ### Comment · Reviewer_KS54 · 2025-08-02
> >
> > Thank you for your explanation. I keep my score as it is.

---

> > > ### Author Response · Authors · 2025-08-03
> > >
> > > Thank you for taking the time to respond. If you have any additional comments or questions—especially suggestions that could help us strengthen the paper—we’d be very grateful to hear them. We’re happy to clarify anything that remains unclear.

---

### Official Review · Reviewer_niBw · 2025-06-21

**Clarity:** 3
**Significance:** 3
**Originality:** 3
**Rating:** 4
**Confidence:** 2

**Summary:**

This paper provides an approximation theory for 1-Lipschitz ResNets. The authors prove that 1-Lipschitz ResNets are dense in 1-Lipschitz continuous functions on compact domains, in two settings: 1) unbounded width and depth; 2) fixed width and unbounded depth.

**Questions:**

No.

**Ethical Concerns:**

["NO or VERY MINOR ethics concerns only"]

**Final Justification:**

The authors' rebuttal addresses most of my concerns. There is theoretical novelty, even if many techniques are from previous works. I would like to recommend "weak accept".

**Limitations:**

Limitations are clearly discussed in Section 5.

**Paper Formatting Concerns:**

No.

**Quality:**

3

**Strengths And Weaknesses:**

__Strengths:__

1. This paper is well-written and is easy to follow.

2. This paper provides an approximation theory for 1-Lipschitz ResNets, which is an important theoretical foundation. The authors establish the theory in two settings: 1) unbounded width and depth; 2) fixed width and unbounded depth.

__Weaknesses:__

1. Though the theoretical analysis is very interesting, most techniques (such as the Restricted Stone-Weierstrass) are already known in the literature, which makes me a bit worried about the novelty.

2. The analysis highly depends on ReLU and is not general enough.

3. Though the authors mention the implementability in Section 5, there are no numerical experiments. This might be fine given the theoretical focus of the paper.

---

> ### Author Rebuttal · Authors · 2025-07-29
>
> Thank you for appreciating our theoretical analysis and for your thoughtful comments. We address your comments below, and we hope you will find the answers satisfying.
>
> ## Weaknesses
>
> ### Novelty of the theoretical derivations
> Most of the techniques we used were indeed developed before our paper. Still, they have never been applied to study the approximation properties of $1$-Lipschitz networks, whose approximation properties were also not understood before our paper. Furthermore, one of the primary novelties in our submission lies in a constructive proof (Sections 3.2 and 4) that does **not rely on the Restricted Stone–Weierstrass theorem**, a central tool in prior works studying this approximation problem. Not directly relying on the Restricted Stone-Weierstrass theorem allowed us to understand that the set of considered networks contains a very well-studied set of functions: 1-Lipschitz piecewise affine functions. This derivation has significant consequences, since it allows us to transfer the approximation properties of this function class to our neural networks. Whereas the Restrictive-Stone Weierstrass theorem provides a very general technique to analyse parametric sets of functions, we also show a constructive strategy to obtain analogous results.
>
> We also believe that the design strategies used for the network in Section 4 are original, and we are not aware of any paper theoretically studying this kind of interaction before. This derivation also brings attention to the fact that to design a $1$-Lipschitz neural network, it is not necessary to entirely rely on $1$-Lipschitz layers. Additionally, while working on this rebuttal, we have noticed another simple yet powerful consequence of our analysis. The set of networks $\widetilde{\mathcal{G}}\_{d,\mathrm{ReLU},d+3}(\mathcal{X},\mathbb{R})$ contains all the piecewise affine $1$-Lipschitz functions, as stated in Theorem 4.2 of our work. On the other hand, it is also true that all the elements of $\widetilde{\mathcal{G}}\_{d,\mathrm{ReLU},d+3}(\mathcal{X},\mathbb{R})$ are piecewise affine $1$-Lipschitz functions. The latter result follows from the fact that $\mathrm{ReLU}$ is piecewise affine, and we only compose it with affine maps. This reasoning allows us to conclude that the new architecture that we propose and study in Theorem 4.1 coincides with the set of piecewise affine $1$-Lipschitz functions, and provides yet another representation strategy for this lattice of functions. We believe this is another original theoretical contribution that strengthens our work. We will include it as a remark in Section 4, and we will make sure that when it is stated that the set of piecewise affine $1$-Lipschitz functions is contained in $\widetilde{\mathcal{G}}\_{d,\mathrm{ReLU},d+3}(\mathcal{X},\mathbb{R})$, we replace it with "it coincides with it".
>
> ### Strong reliance on $\mathrm{ReLU}$
> Thank you for this comment, which was a question posed also by the other reviewers. Even though one can not apply the same techniques to other activation functions, the approximation properties extend to similar neural network architectures. We provide more details on this aspect in our response to Reviewer 6fHT, and we have now written a dedicated appendix that expands on this.
>
> ### Numerical experiments
> Thank you for acknowledging that, being our paper theoretical, it is fine for it not to have numerical experiments. Given the comments of the other reviewers as well, we have also added an appendix dedicated to commenting even further on the practicalities of the implementation of these networks. We have also conducted numerical experiments on a toy problem to demonstrate that our networks have no intrinsic issues with implementability and trainability. We describe the experiments in response to Reviewer 6fHT, and we will include them in a dedicated appendix.

---

> > ### Comment · Reviewer_niBw · 2025-08-07
> >
> > Dear authors,
> >
> > Thank you for your rebuttal. Your answers addressed most of my concerns. There is theoretical novelty, even if many techniques are from previous works. I have updated my rating.

---

> ### Comment · Area_Chair_ZtkX · 2025-08-06
>
> Reviewer niBw,
>
> Could you read the author's rebuttal, update your reviews, and discuss with the authors as needed?
>
> AC

---

### Official Review · Reviewer_cEiW · 2025-06-29

**Clarity:** 3
**Significance:** 3
**Originality:** 3
**Rating:** 5
**Confidence:** 2

**Summary:**

The paper studies approximation capabilities of 1-Lipschitz ResNets (residual networks with a Lipschitz constant c < 1), which are commonly used in inverse problems, generative modeling, and adversarial defense. The main contribution of the paper is the first proof that such models can approximate _any_ 1-Lipschitz function, at least when considering negative gradient steps and interleaving residual layers with affine transformations. Authors first show that this holds under infinite width/depth, but proceed to show that this still holds at a fixed, finite layer width.

**Questions:**

1. On lines 151-152 authors suggest that "the lack of explicit constraint over $\hat{Q}$ leads to problems when implementing these networks". What are these problems? Isn't a lack of constraints easier to implement with gradient-based optimization, i.e. we don't have to resort to projected SGD or other methods?
2. Have authors studied other common activation functions (sigmoid, tanh, etc.), to understand if they have the properties stated on lines 321-322 that would make the proof applicable?

**Ethical Concerns:**

["NO or VERY MINOR ethics concerns only"]

**Final Justification:**

I maintain my recommendation to accept the paper as I believe it to be an important theoretical contribution. I would increase my score  if the paper had a stronger experimental component, helping the reader understand how the proposed architectural changes would impact the model on problems where 1-Lipschitz ResNets are typically applied, like inverse problems, generative modeling, or adversarial defense. While authors have provided numerical results in the rebuttal, the experiments use toy problems.

**Limitations:**

I suggest authors extend their discussion on the potential practical consequences of adding affine layers to a residual architecture.

**Paper Formatting Concerns:**

No concerns.

**Quality:**

4

**Strengths And Weaknesses:**

Strengths:
+ Significant and original theoretical result: the first proof of universality for 1-Lipschitz ResNets, albeit with some deviations from the commonly used architecture (adding affine layers).
+ The proof and prior work are introduced with great clarity, guiding the reader through all the steps involved, their motivation, and how they relate to literature.
+ High-quality manuscript: precise and clear writing, good formatting, consistent mathematical notation.
+ In-depth discussion of the limitations of the proof and their implications.

Weaknesses:
- Authors have to add affine transformations to the ResNets to show universality. While arguably a minor addition, this might have implications in practice, e.g. for optimization. An important benefit of residual networks is better gradient propagation, which might suffer due to the addition of affine layers, especially for deeper nets. Which leads to the next weakness...
- No numerical experiments. While the main contribution of the paper are the universality proofs, authors have to make adjustments to standard architectures to complete the proofs. The contribution would be stronger if authors provided some numerical results to confirm these additions are indeed practical, and don't impact the performance of these models on real problems.

---

> ### Author Rebuttal · Authors · 2025-07-29
>
> Thank you for the thorough analysis of our manuscript, and we are glad for your appreciation of how it is written and the theoretical results we derive. We have prepared an answer for all your comments, and we hope they address your concerns.
>
> ## Weaknesses
> ### Trainability as the depth increases
> All the networks we consider are piecewise affine and $1$-Lipschitz and cannot suffer from exploding gradient problems since the network gradient has $\ell^2$ norm bounded by one. We agree that deep networks could, in principle, suffer from vanishing gradient problems. Still, skip-connections are not the only remedy to vanishing gradient issues, as can be seen from the theoretical studies around the topic of *dynamical isometry*, see references [1] and [2] below. This theory suggests that initialising the network weights so that the input-to-output Jacobian is orthogonal almost everywhere is sufficient to train deep neural networks. This can also be done in our networks, for example, by initialising the intermediate affine maps to the identity, i.e., $A_\ell(x)=x$. The residual layers can also be initialised to have an orthogonal Jacobian almost everywhere (when well defined), by setting $\tau=2$ and initialising $W_\ell$ as an orthogonal matrix. In this way, we get
> \begin{align*}
> \phi_{\theta_\ell}'(x) &= I_h - 2 W_\ell^\top D(x) W_\ell,\\,\\,D(x) = \mathrm{diag}(\mathrm{ReLU}'(W_\ell x + w_\ell)) \\\\
> (\phi_{\theta_\ell}'(x))^\top \phi_{\theta_\ell}'(x) &= I_h + 4(W_\ell^\top D^2(x)W_\ell - W_\ell^\top D(x) W_\ell) = I_h,
> \end{align*}
> and the same holds for $(\phi_{\theta_\ell}'(x)')^\top\phi_{\theta_\ell}'(x)=I_h$, since $D(x)^2=D(x)$ because of the zero and one slopes of $\mathrm{ReLU}$. To demonstrate that there is no intrinsic training problem in these networks with affine layers, we have conducted numerical experiments with networks of varying depth and width. The weights are initialised as described here, so to satisfy the dynamical isometry property. You can find them described in the response to Reviewer 6fHT. Thank you for bringing up this problem, which we now discuss in more detail in our paper when commenting on the practical implementation of our models.
> ### Lack of numerical experiments
> We hope that the additional experimental evidence, together with the explanation in the response to Reviewer 6fHT, can address your concerns.
>
> ## Questions
> ### On lines 151-152
> It is true that it is easier to implement. The problem we wanted to highlight with this statement is that the lack of explicit constraints does not allow us to directly get a $1$-Lipschitz network without regularisation. If one is willing to relax the $1$-Lipschitz requirement during training, then it would be enough to leave $Q$ unconstrained or, at most, introduce some weight decay to control its norm. We have reworded this statement to ensure there is no ambiguity.
> ### Other activation functions
> Thank you for this comment, which was a question posed also by the other reviewers. Even though one can not apply the same techniques to other activation functions, the approximation properties extend to similar neural network architectures. We provide more details on this aspect in our response to Reviewer 6fHT, and we have now written a dedicated appendix that expands on this.
>
> ## Limitations
> ### Practical consequences of using the affine layers
> As recommended, we will extend the discussion of the impact of the affine layers on the network trainability, and support such claims with the numerical experiments discussed above.
>
> ## References used in the response
> - [1] Xiao, L., Bahri, Y., Sohl-Dickstein, J., Schoenholz, S., & Pennington, J. Dynamical Isometry and a Mean Field Theory of CNNs: How to Train 10,000-Layer Vanilla Convolutional Neural Networks. In the International Conference on Machine Learning, 2018. (pp. 5393-5402). PMLR.
> - [2] Saxe, A. M., McClelland, J. L., & Ganguli, S. (2013). Exact solutions to the nonlinear dynamics of learning in deep linear neural networks. In the International Conference on Learning Representations, 2014.

---

> ### Comment · Reviewer_cEiW · 2025-08-07
>
> I thank authors for their detailed response, with numerical results and other clarifications. Numerical results make be more confident that the architectural changes driven by the proof do not have a grave impact on trainability. At the same time, numerical experiments use toy problems (which I consider MNIST to be also): I would love to see the impact of proposed architectural changes in problems where 1-Lipschitz ResNets are typically applied (inverse problems, generative modeling, adversarial defense, etc.), comparing the standard model to the proposed model with interleaved affine layers. Including such results would make for a stronger submission: I keep my score for now, still recommending paper for acceptance.

---

### Official Review · Reviewer_6fHT · 2025-07-03

**Clarity:** 3
**Significance:** 2
**Originality:** 3
**Rating:** 4
**Confidence:** 2

**Summary:**

This paper studies the approximation capabilities of 1-Lipschitz Residual Neural Networks (ResNets) based on explicit Euler steps of negative gradient flows. The authors provide two main theoretical contributions: (1) They prove that 1-Lipschitz ResNets with unbounded width and depth are dense in the set of scalar 1-Lipschitz functions on compact domains, and (2) They show that the same density property holds when the hidden width is fixed by introducing constrained linear maps between residual blocks. The proofs rely on the Restricted Stone-Weierstrass Theorem and constructive arguments showing that these networks can represent all piecewise affine 1-Lipschitz functions.

**Questions:**

This paper makes solid theoretical contributions to understanding 1-Lipschitz ResNets, but falls short of demonstrating practical value. The theoretical results are rigorous and novel, but the complete absence of experimental validation raises serious questions about the practical relevance of these findings.

**Ethical Concerns:**

["NO or VERY MINOR ethics concerns only"]

**Final Justification:**

more experimental results are provided by the authors. I expect to see more convincing results on larger datasets, such as Cifar-10 and Cifar-100.

**Limitations:**

see above

**Quality:**

3

**Strengths And Weaknesses:**

## Strengths

1. Strong theoretical results: The paper provides an universal approximation theory for 1-Lipschitz ResNets.

2. Rigorous mathematical analysis: The dual proof strategies (Restricted Stone-Weierstrass and piecewise affine construction) provide comprehensive theoretical coverage and different perspectives on the problem.


## Weaknesses

1. Lack of experimental validation
The most significant weakness is the **complete absence of any experimental results**. For a neural network paper, this is a critical omission. The authors provide no evidence that:
- Their theoretical constructions are computationally feasible
- The proposed architectures perform competitively with existing methods
- The universal approximation property translates to practical benefits

2. Questionable practical implementability
While the authors claim implementability, several concerns arise:
- The regularization approach for unconstrained Q in G_{d,ρ}(X,R) using terms like Σ(ReLU(||∇_x N_θ(x_i)||_2 - 1))^2 is **not friendly for unconstrained optimization of deep neural networks**
- Spectral norm constraints require expensive power method computations during training
- Row-wise ℓ_2 norm constraints add significant computational overhead
- No analysis of computational complexity or training stability

However, these operation are not friendly for the optimization of deep neural network

### 3. Limited scope and extensions
- Results are restricted to scalar functions (c=1 case), with vector-valued extensions only briefly mentioned
- Heavy reliance on ReLU activation limits generalizability, practically, we use silu, swiglu and etc.



## Minor Issues

I have one suggestion. Adopt standard mathematical typography conventions:
- Use **bold** for vectors (e.g., **x**, **v**)
- Use regular font for scalars (e.g., L, h, d)
- Use **uppercase bold** for matrices (e.g., **W**, **Q**, **A**)
This would improve readability and help readers quickly distinguish between different mathematical objects.

---

> ### Author Rebuttal · Authors · 2025-07-29
>
> Thank you for judging our theoretical contributions as strong and for carefully going through the manuscript. We include below our responses to your comments, and we hope they will make you consider increasing the score.
>
> ## Weaknesses
> ### Lack of experimental validation.
> This submission is a theoretical paper, and our main contribution lies in the development of novel proof techniques and in providing two alternative viewpoints to our analysis, thereby gaining a deeper understanding of $1$-Lipschitz ResNets. Not directly relying on the Restricted Stone-Weierstrass theorem allowed us to understand that the set of considered networks contains the set of 1-Lipschitz piecewise affine functions, hence allowing us to transfer their approximation properties to our networks. This proof is also constructive, unlike those relying on the Restricted Stone-Weierstrass Theorem.
>
> We also introduce a new neural network architecture, in Section 4, which theoretically displays desirable expressiveness.
>
> We now support our claims regarding the implementability of these models with some new numerical experiments, as described in the response to the next question, which we will include in an appendix.
>
> ### Questionable practical implementability.
> The regularisation strategy we mention in the paper is one of the most commonly adopted choices for the training of the critic in Wasserstein generative models, hence demonstrating its practical value and implementability.
>
> We agree that the power method could be costly. Still, what is commonly done and turns it into an efficient solution, is to do several power iterations at the initialisation to have an accurate approximation of the leading singular vector, and then doing only one per gradient step, since the weights tend to change very mildly during training, and those pre-computed singular vectors are very good initial guesses. This is discussed in references [1] and [2] below.  As shown in the experiments below, the cost of this step is negligible compared to the overall cost of the forward and backwards passes.
>
> We agree that evaluating the row/block constraints could be expensive following the procedure we described. Such a procedure could still be efficiently replaced by a power iteration method. Thank you for pointing out these potential computational limitations. We have prepared an additional appendix that comments on these aspects, thereby justifying the practical implementability of our models.
>
> In addition, we have conducted numerical tests to demonstrate that there is no inherent problem in training the considered networks, even with increasing network depth. We consider the task of classifying the points of the planar two-moon dataset with additive Gaussian noise (standard deviation=0.1). The network weights are initialised to satisfy the dynamical isometry property, as described in the response to Reviewer cEiW.
>
> All the layers are constrained as in the paper, but the first one is left unconstrained to have fair comparisons between the two types of models. We also present the average time required to implement the constraints over epochs measured in seconds via Python's *time* library. The code is written in PyTorch and will be made available upon acceptance.
>
> Both models consistently train regardless of the network width and depth. The cost of constraining them is larger for those with affine layers, and it grows faster as the depth increases than when the width does. Still, we believe the implementation could be optimised, and this step could be made faster. In Tables 1 and 2, we denote with Theorem 3.1 the architecture without affine layers, and with Theorem 4.1, with affine layers and further constraints on the gradient steps.
>
> **Table 1: Networks with varying depth and fixed width.** Numerical experiments for the two-moon dataset with $\sigma=0.1$ as standard deviation of the additive Gaussian noise. The hidden dimension is set to $d+3=5$, where $d=2$ is the input dimension of the data points.
>
> | **Number layers** | **Theorem 3.1** |  |  **Theorem 4.1** | |
> |:--------------|------------------------:|:-------------------|------------------------:|:-------------------|
> |  | **Test Accuracy** | **Normalisation Time** | **Test Accuracy** | **Normalisation Time** |
> | L = 2  | 99.75 % | 0.0079 | 88.25 % | 0.053 |
> | L = 4  | 99.88 % | 0.0130 | 99.88 % | 0.097 |
> | L = 8  | 100.00 % | 0.0286 | 99.88 % | 0.185 |
> | L = 16 | 100.00 % | 0.0585 | 100.00 % | 0.273 |
> | L = 32 | 99.88 % | 0.1091 | 100.00 % | 0.326 |
> | L = 64 | 100.00 % | 0.1628 | 100.00 % | 0.685 |
>
>
> **Table 2: Networks with fixed depth and varying width.** Numerical experiments for the two-moon dataset with $\sigma=0.1$ as standard deviation of the additive Gaussian noise. The number of layers is fixed to $L=10$, and the hidden dimension is varied.
>
> | **Network width** | **Theorem 3.1** |  |  **Theorem 4.1** | |
> |:--------------|------------------------:|:-------------------|------------------------:|:-------------------|
> |  | **Test Accuracy** | **Normalisation Time** | **Test Accuracy** | **Normalisation Time** |
> | h = 10 | 99.88 % | 0.0372 | 100.00 % | 0.1628 |
> | h = 20 | 99.88 % | 0.0379 | 99.88 % | 0.1744 |
> | h = 30 | 99.88 % | 0.0381 | 99.88 % | 0.1829 |
> | h = 40 | 100.00 % | 0.0378 | 99.63 % | 0.2993 |
>
>
> We will include an appendix describing these experiments and the obtained results, which corroborate the theory and support our comments on the practical implementability of the models.
>
> ### Limited scope and extensions.
> We agree that focusing only on the scalar case is a limitation of our contribution. Still, the vector-valued case is much more challenging, and we believe that it cannot be done before the details for the scalar case are fixed, which is the purpose of our manuscript. The focus on scalar functions is common among the papers studying similar networks, see [3] and [4] below. We show the complexities of the extension when stating and commenting Lemma 5.1.
>
> The focus on a specific activation, and in particular on $\mathrm{ReLU}$, is quite common in the mathematics of deep learning, as can be seen, for example, in these papers [5], [6], and [7] below. For any other continuous activation $\sigma:\mathbb{R}\to\mathbb{R}$ which is not a polynomial, it is relatively immediate to recover the density of the set of networks $\mathcal{G}\_{d,\mathcal{N}\_\sigma}(\mathcal{X},\mathbb{R})$ in $\mathcal{C}\_1(\mathcal{X},\mathbb{R})$, where $\mathcal{N}\_\sigma:\mathbb{R}\to\mathbb{R}$, $\mathcal{N}\_\sigma(s) = u\_1^\top \sigma(u_2 s + u_3)$ with $u\_1,u\_2,u\_3\in\mathbb{R}^h$, is a sufficiently wide single hidden layer network based on $\sigma$. To prove this, we can approximate the elements in $\mathcal{G}\_{d,\mathrm{ReLU}}(\mathcal{X},\mathbb{R})$ to an arbitrary accuracy. This holds because the residual layers in $\mathcal{G}\_{d,\mathrm{ReLU}}(\mathcal{X},\mathbb{R})$ are gradient steps for potential energies of the form $g\_\ell(x)=1\_{h\_{\ell}}^\top \mathrm{ReLU}^2(W\_\ell x + b\_\ell)/2$ and, on compact sets, we can simultaneously approximate $\mathrm{ReLU}$ and $\mathrm{ReLU}^2/2$ to an arbitrary accuracy with a sufficiently wide single hidden layer network based on a continuously differentiable non-polynomial activation function $\gamma$, see Theorem 4.1 in [8]. We can then call $\gamma'=\sigma$. In this case, though, the imposition of the $1$-Lipschitz constraints in practice becomes even more challenging, since we wouldn't be able to implement residual steps which are $1$-Lipschitz directly, given that it is not known if $\mathcal{N}\_{\sigma}$ is $1$-Lipschitz and monotonically increasing. These networks would generally be much more expensive to evaluate than $\mathrm{ReLU}$ ones. Moreover, the core results are still those for the $\mathrm{ReLU}$ function in this case. A particularly simple example is $\sigma(x)=\mathrm{LeakyReLU}\_{\alpha}(x)=\max\\{\alpha x, x\\}$, $\alpha \in (0,1)$ where we do not need approximations since
> $$
> \mathrm{ReLU}(x) = \frac{1}{1-\alpha^2}\left(\mathrm{LeakyReLU}\_{\alpha}(x) + \alpha \mathrm{LeakyReLU}\_{\alpha}(-x)\right).
> $$
> We now expand on these generalisations in a dedicated appendix.
>
> ## Minor issues
> ### Bold notation for vectors.
> Thank you for your suggestion. We prefer sticking to the current notation, as the pages would become too heavy in our opinion.
>
> ## Questions
> ### Solid theoretical contributions, but falls short of demonstrating practical value.
> We hope that the numerical experiments we have now conducted can address your concerns, complementing and corroborating our theoretical analysis.
>
> ## References
> - [1] Miyato, T., Kataoka, T., Koyama, M., & Yoshida, Y. (2018). Spectral Normalization for Generative Adversarial Networks. International Conference on Learning Representations.
> - [2] Sherry, F., Celledoni, E., Ehrhardt, M. J., Murari, D., Owren, B., & Schönlieb, C. B. (2024). Designing stable neural networks using convex analysis and ODEs. Physica D: Nonlinear Phenomena, 463, 134159.
> - [3] Neumayer, S., Goujon, A., Bohra, P., & Unser, M. (2023). Approximation of Lipschitz Functions Using Deep Spline Neural Networks. SIAM Journal on Mathematics of Data Science, 5(2), 306-322.
> - [4] Cem Anil, James Lucas, and Roger Grosse. Sorting Out Lipschitz Function Approximation. In the International Conference on Machine Learning, pages 291–301. PMLR, 2019.
> - [5] Yarotsky, D. (2018, July). Optimal approximation of continuous functions by very deep ReLU networks. In Conference on learning theory (pp. 639-649). PMLR.
> - [6] Opschoor, J. A., Petersen, P. C., & Schwab, C. (2020). Deep ReLU networks and high-order finite element methods. Analysis and Applications, 18(05), 715-770.
> - [7] Hanin, B., & Sellke, M. (2017). Approximating Continuous Functions by ReLU Nets of Minimal Width. arXiv preprint arXiv:1710.11278.
> - [8] Pinkus, A. (1999). Approximation theory of the MLP model in neural networks. Acta numerica, 8, 143-195.

---

> > ### Comment · Reviewer_6fHT · 2025-08-04
> > **Official Comment by Reviewer 6fHT**
> >
> > Thank you very much for your reply. I still have concern about "Lack of experimental validation.".
> >
> > The two-moon dataset seems to be a toy dataset. Can you please verify Theorem 3.1 and Theorem 4.1 on the MNIST or the CIFAR-10 dataset?

---

> > > ### Author Response · Authors · 2025-08-05
> > > **MNIST experiments**
> > >
> > > Thank you for your response.
> > >
> > > We have run some experiments with the architecture in Theorem 4.1 trained to classify the MNIST dataset. The one in Theorem 3.1 is not new, and our paper focuses on studying its properties. This means that there is already strong evidence that such a network can be trained, and it was trained to classify several datasets such as CIFAR-10 and CIFAR-100, see for example Table 2 in [1], Figure 1 in [2], and Table 2 and Figure 5 in [3]. We mention in particular [3] where the authors compare several $1$-Lipschitz networks and, in the conclusions, they say that "...the results are in favor of CPL, due to its highest performance and lower consumption of computational resources. ", where CPL stands exactly for the model in Theorem 3.1.
> > >
> > > The use of the architecture of Theorem 4.1 in the context of convolutional neural networks is still premature, since we have not yet extended the affine layers and their constraints to convolutional layers, and this is not our focus in this paper. We therefore focus on MNIST, which can be accurately classified with a fully connected network.
> > >
> > > We compare nine architectures with varying width and depth. The grey-scale images of MNIST have 784 pixels. Since preserving or increasing such a dimensionality would lead to an excessively large network for the task at hand, we reduce the dimension to $d\in \\{50, 100, 200\\}$ with a first affine layer. We then apply the architecture in Theorem 4.1, where the hidden dimension is $h=d+3\in\\{53, 103,203\\}$. We consider architectures with $L\in\\{5,10,20\\}$ layers and report the validation accuracies in the table below.
> > >
> > > **Results MNIST:** Network defined based on Theorem 4.1 and trained to classify the MNIST dataset. We compare different network widths and depths.
> > > |  | $L=5$ | $L=10$| $L=20$ |
> > > | :--------: | :-------: | :-------: | :-------: |
> > > | $d=50$  | 97.85%  | 97.67% | 97.82% |
> > > | $d=100$ | 97.94% | 97.70% | 97.58% |
> > > | $d=200$    | 97.68%   | 97.77%  | 97.89% |
> > >
> > > The models all perform similarly accurately in this classification task. As for the two-moon dataset experiments, we do not constrain the first linear lifting/resizing layers, but we constrain all the others as described in the paper. We remark that this task does not entirely represent our theoretical results, since we focus on approximating scalar-valued functions. In contrast, in this case we have outputs in $\mathbb{R}^{10}$, given that $10$ is the number of classes in the MNIST dataset. For this reason, we do not constrain the last affine layer either.
> > >
> > > The value of these experiments lies in strengthening the claims about the trainability of these newly proposed models, which, despite the addition of interleaved affine layers, do not suffer from practical problems.
> > >
> > > We hope that these experiments on a more complicated dataset address your concern about "Lack of experimental validation" and may make you consider increasing the score.
> > >
> > > ### References
> > > - [1] Meunier, L., Delattre, B. J., Araujo, A., & Allauzen, A. (2022, June). A Dynamical System Perspective for Lipschitz Neural Networks. In International Conference on Machine Learning (pp. 15484-15500). PMLR.
> > > - [2] Sherry, F., Celledoni, E., Ehrhardt, M. J., Murari, D., Owren, B., & Schönlieb, C. B. (2024). Designing stable neural networks using convex analysis and ODEs. Physica D: Nonlinear Phenomena, 463, 134159.
> > > - [3] Prach, B., Brau, F., Buttazzo, G., & Lampert, C. H. (2024). 1-Lipschitz Layers Compared: Memory, Speed, and Certifiable Robustness. In Proceedings of the IEEE/CVF Conference on Computer Vision and Pattern Recognition (pp. 24574-24583).

---

> > > > ### Comment · Reviewer_6fHT · 2025-08-08
> > > > **Thank you for your reply.**
> > > >
> > > > Thank you to the authors for their detailed response, especially for providing numerical results, which make the work more practical. However, the current experiments are conducted only on the MNIST dataset, which is a toy dataset. Even a simple two-layer MLP can achieve high performance on it, so the effectiveness of the proposed method is not particularly surprising. Although I will increase my score to 4, I am not fully confident in this assessment. Similar to Reviewer cEiW, I would expect to see stronger results on at least CIFAR-10 or CIFAR-100 in a revised submission.

---

### Official Review · Reviewer_2dAP · 2025-07-06

**Clarity:** 2
**Significance:** 3
**Originality:** 3
**Rating:** 4
**Confidence:** 3

**Summary:**

This paper studies the approximation capabilities of 1-Lipschitz ResNets, with ReLU activation function. It proves that 1-Lipschitz ResNets with unbounded width and length can approximate any 1-Lipschitz scalar function. The key to prove the results is to construct a 1-Lipschitz ResNets layer, which the paper leverages the existing results (called the explicit Euler step for the negative gradient flow differential equation; see Proposition 2.1 in line 125 and the remarks right after). The paper further proves that the same approximation capability holds when the width is fixed (specifically, width length can be fixed to be no less than d+3 where d is the input dimension) and only the depth is unbounded. The proof builds upon an understanding that any piecewise affine 1-Lipschitz scalar function can be presented by a fixed width 1-Lipschitz ResNets; the result is interesting. As a theory paper, there is no numerical experiments to validate or show the usefulness of the results under practical settings.

**Questions:**

1. Perhaps the authors could give more concrete motivation of why focusing on 1-Lipschitz NN and why addressing the research gap, i.e., proving the 1-Lipschitz ResNets can approximate any 1-Lipschitz function is important. For example, why just ResNets but not other NN architectures like GNN, Transformers, etc? Can the techniques for ResNets be applicable to GNN /Transformers?
2. Perhaps the authors could highlight the novelty of the techniques used in the proof and the significance of the results (by for example putting them with respect to the existing universal approximation results for ResNets).

I am ready to increase my score if the above issues are well addressed.

**Ethical Concerns:**

["NO or VERY MINOR ethics concerns only"]

**Final Justification:**

The response has addressed my concerns on the motiviation of the study, the challenges, and the novelty. I think this is a good theory paper with notable contributions and should be accepted.

**Limitations:**

The paper presents discussions on some proof techniques, extension of the results, and implementation of the proposed ResNets architecture in Sec. 5.

**Paper Formatting Concerns:**

N.A.

**Quality:**

3

**Strengths And Weaknesses:**

Strengths:
- The paper appears to address a (less-motivated) research gap that 1-Lipschitz ResNets can approximate any 1-Lipschitz scalar function arbitrarily well.
- The authors are knowledgeable about the relevant techniques and leverage them towards proving the results in this paper.
- The paper is relatively easy to follow.

Weaknesses:

- The motivation can be more concrete and stronger. The paper only briefly states the importance 1-Lipschitz NN without giving concrete evidence. Same for the claim in the introduction that “The commonly adopted strategies to constrain a network’s Lipschitz constant tend to reduce its expressiveness …”. Concrete evidence would strengthen the motivation, especially for laymen.
- The challenges in proving the main results and their significance, that 1-Lipschitz ResNets with unbounded width and depth (or bounded width and unbounded depth) can approximate any 1-Lipschitz scalar function, should be discussed. It will be great if the authors can highlight the significance of such results in a broader context (beyond that it addresses a weekly-motivated research gap).

Minor comments:

- Why presenting Theorem 3.1, if the results in Theorem 4.1 already covers those in Theorem 3.1?  (Thus, one can present Theorem 4.1 directly, which seems to involve more new developments in this paper anyway.)

---

> ### Author Rebuttal · Authors · 2025-07-29
>
> Thank you for carefully reading our paper and explicitly stating your interest in raising the score in case we address your questions. We report below our responses. We collect all the mentioned references at the end of the response.
>
> ## Weaknesses
>
> ### Motivation
> Thank you for this suggestion on strengthening the motivation behind the paper and the necessity to study how to design expressive $1$-Lipschitz neural networks. In the response to the next question, we provide a reference on the impossibility of approximating $1$-Lipschitz functions with a commonly used type of $1$-Lipschitz neural networks. This motivates the need to study and design less conventional architectures. Another important reason to study these models stems from experimental results in Figure 1 of [8] and similar findings in other related papers, where the authors train Lipschitz-continuous gradient-based ResNets and observe improved robustness to adversarial attacks, accompanied by a drop in performance on clean images. We believe that understanding the theoretical expressive power of these models is crucial for determining whether the experimental results arise from intrinsic architectural limitations or the training methods, as our theory suggests. When revising the manuscript, we will expand on these two aspects and also provide a longer description of the other references that we have already included.
>
> Regarding the importance of $1$-Lipschitz neural networks, we have already cited some works relying on them in the related work section. We will expand on how essential the $1$-Lipschitz constraint is in those contexts. To do so, we will also more explicitly describe two problems where $1$-Lipschitz constraints are desirable in an appendix. These two problems are:
> 1. Improving the network robustness to adversarial attacks, see for example [4], [8], and [11],
> 2. Deploying a network trained as an image denoiser in a Plug-and-Play algorithm for image reconstruction in inverse problems, for example, for deblurring problems, see [8], [9], and [10].
>
> In general, also in all those situations where one is interested in learning maps that are then iteratively applied, aiming to converge to a fixed point, it is possible to guarantee the convergence of the iterative scheme only under some Lipschitz constraints.
>
> ### Significance of the results
> We tend to disagree in saying that the results are hardly surprising, in the sense that, even though unconstrained ResNets can approximate an arbitrary continuous function, it is not necessarily the case that the same holds for a constrained one. For example, in Proposition 3.3 of reference [1] below, the authors prove that $\mathrm{ReLU}$-feedforward networks with unit-norm weights, which are $1$-Lipschitz, are not dense in the set of $1$-Lipschitz functions. This holds despite the unconstrained models being universal approximators of continuous functions. Given your comments, we agree that it is very important to remark the relevance of our contribution and of the study of the approximation properties of $1$-Lipschitz ResNets. We will do so in the revised manuscript. Thank you.
>
> ## Minor comments
> ### Theorem 4.1 provides a generalisation of Theorem 3.1.
> Theorem 4.1 is not a generalisation of Theorem 3.1, strictly speaking, since for Theorem 4.1 to hold we need to introduce the affine layers, whereas Theorem 3.1 only relies on networks with residual layers. We comment on this aspect in the paragraph starting on line 281.
>
> ## Questions
> ### Motivation behind our work and why did we study ResNets
> There are three main reasons why we focus on ResNets:
> 1. $1$-Lipschitz constrained ResNets have been used extensively in several applications;
> 2. A theory similar to ours for feedforward networks has already been developed;
> 3. GNNs, Transformers, and other architectures also rely on residual connections which are hence an essential piece to understand.
>
> We agree that providing additional context and defining the research gap we are addressing would strengthen the paper. We have now expanded on this aspect in the introduction, also stressing the original theoretical ideas we introduced and their transferability. We mention in particular GNNs that have already been considered in the same gradient form before (see, for example, references [2], [3] and [4] below). While we are not aware of the use of these gradient layers in Transformers, there has already been interest in studying Lipschitz-continuous transformers, see [5], [6], and [7] below, and we believe that our theoretical analysis could at least partially translate to these more complex architectures.
>
> ### Novelty of the techniques used in the proof and the significance of the results
> Regarding the novelty of our theoretical derivations, we would like to clarify that our main contribution lies in proving the approximation theorem without relying on the Restricted Stone-Weierstrass theorem (see Sections 3.2 and 4), and in providing two alternative viewpoints to our analysis, thereby gaining a deeper understanding of $1$-Lipschitz ResNets. Not directly relying on the Restricted Stone-Weierstrass theorem forced us to understand that the set of considered networks contains a very well-studied set of functions: 1-Lipschitz piecewise affine functions. This derivation has significant consequences, since it allows us to transfer the approximation properties of this function class to our neural networks. Whereas the Restrictive-Stone Weierstrass theorem provides a very general technique to analyse parametric sets of functions, we also show a constructive strategy to obtain analogous results.
>
> We also introduce a new neural network architecture in Section 4, which theoretically exhibits desirable expressiveness and has not been explored before. Additionally, while working on this rebuttal, we have noticed another simple yet powerful consequence of our analysis. The set of networks $\widetilde{\mathcal{G}}\_{d,\mathrm{ReLU},d+3}(\mathcal{X},\mathbb{R})$ contains all the piecewise affine $1$-Lipschitz functions, as stated in Theorem 4.2 of our work. On the other hand, it is also true that all the elements of $\widetilde{\mathcal{G}}\_{d,\mathrm{ReLU},d+3}(\mathcal{X},\mathbb{R})$ are piecewise affine $1$-Lipschitz functions. The latter result follows from the fact that $\mathrm{ReLU}$ is piecewise affine, and we only compose it with affine maps. This reasoning allows us to conclude that the new architecture that we propose and study in Theorem 4.1 coincides with the set of piecewise affine $1$-Lipschitz functions, and provides another representation strategy for this lattice of functions. We believe this is another original theoretical contribution that strengthens our work. We will include it as a remark in Section 4 and make sure that whenever we say that such functions are contained in the parametric set $\widetilde{\mathcal{G}}\_{d,\mathrm{ReLU},d+3}(\mathcal{X},\mathbb{R})$, we replace it with "coincides with it".
>
> We will ensure that the relevance of these contributions is emphasised more in the paper. Given the comments of some other reviewers, we have also performed simple numerical experiments to demonstrate the performance of these models, as seen in the responses to Reviewer 6fHT.
>
> ## References
> - [1] Neumayer, S., Goujon, A., Bohra, P., & Unser, M. (2023). Approximation of Lipschitz Functions Using Deep Spline Neural Networks. SIAM Journal on Mathematics of Data Science, 5(2), 306-322.
> - [2] Chen, Q., Wang, Y., Wang, Y., Yang, J., & Lin, Z. (2022, June). Optimization-Induced Graph Implicit Nonlinear Diffusion. In International Conference on Machine Learning (pp. 3648-3661). PMLR.
> - [3] Eliasof, M., Haber, E., & Treister, E. (2021). PDE-GCN: Novel Architectures for Graph Neural Networks Motivated by Partial Differential Equations. Advances in Neural Information Processing Systems, 34, 3836-3849.
> - [4] Eliasof, M., Murari, D., Sherry, F., & Schönlieb, C. B. (2024). Resilient Graph Neural Networks: A Coupled Dynamical Systems Approach. In ECAI 2024 (pp. 1607-1614). IOS Press.
> - [5] Castin, V., Ablin, P., & Peyré, G. (2023). How Smooth Is Attention?. arXiv preprint arXiv:2312.14820.
> - [6] Kim, H., Papamakarios, G., & Mnih, A. (2021, July). The Lipschitz Constant of Self-Attention. In International Conference on Machine Learning (pp. 5562-5571). PMLR.
> - [7] Qi, X., Wang, J., Chen, Y., Shi, Y., & Zhang, L. (2023). LipsFormer: Introducing Lipschitz Continuity to Vision Transformers. arXiv preprint arXiv:2304.09856.
> - [8] Sherry, F., Celledoni, E., Ehrhardt, M. J., Murari, D., Owren, B., & Schönlieb, C. B. (2024). Designing stable neural networks using convex analysis and ODEs. Physica D: Nonlinear Phenomena, 463, 134159.
> - [9] Cheng, Y., Schönlieb, C. B., & Aviles-Rivero, A. I. (2024). You KAN Do It in a Single Shot: Plug-and-Play Methods with Single-Instance Priors. arXiv preprint arXiv:2412.06204.
> - [10] Hertrich, J., Neumayer, S., & Steidl, G. (2021). Convolutional Proximal Neural Networks and Plug-and-Play Algorithms. Linear Algebra and its Applications, 631, 203-234.
> - [11] Prach, B., Brau, F., Buttazzo, G., & Lampert, C. H. (2024). 1-Lipschitz Layers Compared: Memory, Speed, and Certifiable Robustness. In Proceedings of the IEEE/CVF Conference on Computer Vision and Pattern Recognition (pp. 24574-24583).

---

> > ### Comment · Reviewer_2dAP · 2025-08-02
> > **Response to rebuttal**
> >
> > Thanks for the detailed and helpful response. It addressed my concerns and I will raise my score accordingly. It would be nice to see some of the helpful discussions in the response in the revised version of the paper.

---

> > > ### Author Response · Authors · 2025-08-03
> > >
> > > Thank you for the response and for increasing your score.

---

### Decision · Program_Chairs · 2025-09-17

**Decision:**

Accept (poster)

**Comment:**

The  paper studies the approximation capabilities of 1-Lipschitz ResNets and provides the first universal approximation guarantees for 1-Lipschitz ResNets with ReLU activation, under unbounded width and depth, or fixed width but with a modified architecture. This paper provides strong theoretical results and rigorous mathematical analysis. The choice of 1-Lipschitz ResNets with ReLU is well-motivated, according to the author's responses added during the rebuttal, given the effectiveness of 1-Lipschitz ResNets in improving robustness and the use of residual connections in broader models such as Transformers. Additionally, ReLU is used, which is a more commonly used activation function than other activation functions such as GroupSort that previous works had to use. The paper has some novel contributions on the proof techniques which do not use the Stone-Weierstrass theorem in previous works, but the significance of novelty looks kind of limited, as it sounds like similar results can also be obtained by applying existing techniques with the Stone-Weierstrass theorem. There are common concerns on the practical implementability, and the paper is almost purely theoretical and lacks experiments for actual demonstration; some experiments have been added during the rebuttal but are still kind of toy; though this can be normal considering this paper focuses on the theory. Overall, the theoretical contributions are still strong and all reviewers gave positive ratings.